# AsyncMesh: Fully Asynchronous Optimization for Data and Pipeline Parallelism

## Abstract

Data and pipeline parallelism are key strategies for scaling neural network training across distributed devices, but their high communication cost necessitates co-located computing clusters with fast interconnects, limiting their scalability. We address this communication bottleneck by introducing *asynchronous updates across both parallelism axes*, relaxing the co-location requirement at the expense of introducing *staleness* between pipeline stages and data parallel replicas. To mitigate staleness, for pipeline parallelism, we adopt a weight look-ahead approach, and for data parallelism, we introduce an *asynchronous sparse averaging* method equipped with an exponential moving average based correction mechanism. We provide convergence guarantees for both sparse averaging and asynchronous updates. Experiments on large-scale language models (up to *1B parameters*) demonstrate that our approach matches the performance of the fully synchronous baseline, while significantly reducing communication overhead.

## 1 Introduction

Distributed optimization approaches enable large-scale model training by partitioning computation across multiple interconnected devices, primarily through Data Parallelism (DP) (Goyal, 2017; Li et al., 2020; Zhao et al., 2023) and Model Parallelism (MP) (Huang et al., 2019; Krizhevsky et al., 2017; Shoeybi et al., 2019). While DP replicates the model across devices with partitioned data, MP partitions the model itself across devices. Combining these approaches allows training foundation models at the frontier scale (Dubey et al., 2024; Liu et al., 2024a). However, both DP and MP rely on high-bandwidth interconnects due to high communication costs, limiting distributed training to co-located computing clusters. Therefore, scaling beyond a centralized infrastructure requires addressing communication bottlenecks inherent to both DP and MP setups.

Existing approaches mitigate communication costs via compression (Bernstein et al., 2018; Wang et al., 2023; Wangni et al., 2018) and/or overlapping computation with communication using scheduling (Narayanan et al., 2019; Qi et al., 2023) or asynchronous methods (Agarwal & Duchi, 2011; Stich & Karimireddy, 2019). We focus on asynchronous methods, which by design, offer full utilization of the distributed infrastructure and support heterogeneous hardware, by eliminating synchronization barriers. In this work, we study asynchronous training in a *2D mesh* combining DP and Pipeline Parallelism (PP) (Huang et al., 2019) – a special case of MP that partitions the model into sequential stages.

In a synchronous mesh, each stage within a pipe[1] is optimized in a lock-step manner, ensuring that the weights and gradients are synchronized at each step, and the model replicas are explicitly averaged by pausing optimization. In contrast, **AsyncMesh** eliminates both synchronization points allowing decoupled optimization in each stage as well as decoupling optimization in a pipe and averaging across replicas. This enables *continuous data processing without communication bottlenecks*. Fig. 1 for an illustrates this. Asynchronism, however, introduces *staleness* in model weights, necessitating correction mechanisms to ensure consensus among model replicas and convergence (Agarwal & Duchi, 2011; Ajanthan et al., 2025a; Stich & Karimireddy, 2019; Zheng et al., 2017).

For PP, the staleness is stage-dependent (Narayanan et al., 2019), and extrapolating the previous update direction using the Nesterov method is shown to effectively compensate for this delay (Ajanthan

---

[1]A pipe is a sequence of stages that form a full model.

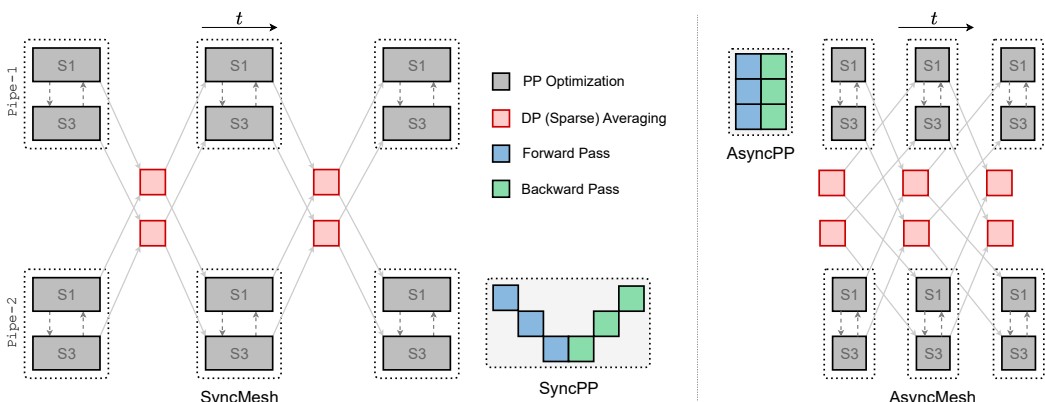

Figure 1: *SyncMesh vs AsyncMesh, for a 3-stage, 2-DP replica setup (only 2 stages: S1 and S3, are shown for clarity). Notably, AsyncMesh eliminates idle time due to communication for both PP and DP. In synchronous DP, devices are idle while parameters are averaged, whereas asynchronous DP eliminates this idle time by using the "old" average. Moreover, in AsyncPP, each stage alternates between forward and backward passes without any communication delay.*

et al., 2025a). Due to its simplicity and empirical effectiveness, we adopt this strategy (Ajanthan et al., 2025a) to optimize each pipe asynchronously. For DP, the staleness depends on the interconnect bandwidth and the data transfer volume. Therefore, to minimize staleness, we *exchange only 5% of weights* across replicas, similar to sparse averaging (Beton et al., 2025; Fournier et al., 2024), and communicate them asynchronously to *completely mask the DP communication*. To address resulting weight discrepancies, we design an *Exponential Moving Average (EMA)* based correction mechanism that approximates the average staleness. Theoretical analysis under the homogeneous setting (identical hyperparameters and i.i.d. data splits) shows that our method ensures consensus among replicas on expectation, despite asynchronous sparse averaging.

We validate our approach on large-scale language modeling tasks with decoder-only transformer architectures (Karpathy, 2022; Vaswani et al., 2017). Experiments demonstrate that our method *matches the performance of the fully synchronous baseline*, while eliminating the communication overhead via sparse asynchronous updates. Notably, for the first time, we train a *1B parameter model* to convergence in AsyncMesh matching the performance of the synchronous alternative. Our results show the feasibility of distributed training over bandwidth constrained interconnects using asynchronous optimization.

Our contributions can be summarized as follows:

- We introduce a new **AsyncMesh** setup where *both DP and PP are asynchronous*, and present a fully asynchronous method that can match the performance of the fully synchronous method.
- We provide theoretical justification of convergence in the presence of staleness in a homogeneous setup where only a small subset of weights is communicated between DP replicas.
- We empirically demonstrate the robustness and scalability of our approach across varying subset sizes, staleness levels, DP communication intervals, and PP $\times$ DP mesh sizes.

## 2 PRELIMINARIES

We first define the 2D mesh configuration with DP (Goyal, 2017; Li et al., 2020) and PP (Huang et al., 2019), and then briefly review the Asynchronous Pipeline Parallel (AsyncPP) method (Ajanthan et al., 2025a) and Sparse Parameter Averaging (SPARTA) (Beton et al., 2025; Fournier et al., 2024) upon which we build our work. We refer the reader to the respective papers for more details.

### 2.1 PROBLEM SETUP: 2D MESH

Let $P$ be the number of pipeline stages and $m$ be the number of replicas. We consider a symmetric 2D mesh, where each stage is replicated $m$ times, constituting $Pm$ devices (or workers) in total. A *pipe* is a sequence of stages that form the full model, which can be defined using its forward and

backward functions. Consider a pipe (or replica) $i \in \{1, \ldots, m\}$, and let the forward function for stage $j$ be $f_{ij} := f_j(\mathbf{w}_{ij}; \mathbf{x}_{j-1})$ with weights $\mathbf{w}_{ij}$, and input $\mathbf{x}_{j-1}$. Then, the forward and backward functions for pipe $i$ can be defined as:

$$F(\mathbf{W}_i; \mathbf{x}_0) := f_{iP} \circ f_{iP-1} \circ \cdots \circ f_{i1}(\mathbf{x}_0) , \qquad \text{forward} , \qquad (1)$$
$$G(\mathbf{W}_i; \mathbf{e}_P) := g_{i1} \circ g_{i2} \circ \cdots \circ g_{iP}(\mathbf{e}_P) , \qquad \text{backward} ,$$

where $\mathbf{W}_i = \{\mathbf{w}_{iP}, \ldots, \mathbf{w}_{i1}\}$ and $\mathbf{x}_0$ is an input data point. Here, $g_{ij} := g_j(\mathbf{w}_{ij}; \mathbf{e}_j)$ is the backward function for stage $j$ corresponding to $f_{ij}$ and $\mathbf{e}_P$ is the error signal corresponding to $\mathbf{x}_0$. In our 2D mesh, there are $m$ such pipes, and the goal is to optimize the following consensus objective (Boyd et al., 2011):

$$\min_{\mathbf{W} \in \mathbb{R}^d} F(\mathbf{W}; \mathcal{D}) := \min_{\mathbf{W}_i \in \mathbb{R}^d} \sum_{i=1}^{m} F(\mathbf{W}_i; \mathcal{D}_i) , \qquad (2)$$
$$\text{s.t.} \quad \mathbf{W}_i = \mathbf{W} , \qquad \forall i \in \{1, \ldots, m\} ,$$

where $\mathcal{D}_i$ is an i.i.d. subset of the dataset $\mathcal{D}$ and $d$ is the number of learnable parameters of the full model. Each pipe is optimized independently on its own data split and using its own optimizer, and its weights are synchronized periodically with other pipes – typically after every optimization step.

## 2.2 PIPELINE PARALLELISM

Pipeline Parallelism (PP) methods (Guan et al., 2024) provide communication efficient ways to optimize the model parameters in a pipe. Specifically, pipeline scheduling strategies such as GPipe (Huang et al., 2019), 1F1B (Narayanan et al., 2021b), and ZeroBubble (Qi et al., 2023) design the order of processing forward and backward passes of microbatches in a pipe to reduce communication overhead between stages and improve device utilization. In contrast, asynchronous PP methods (Ajanthan et al., 2025a; Narayanan et al., 2019) eliminate the requirement to synchronize the weights and gradients across stages at each update step, offering full pipeline utilization.[2]

**Asynchronous Pipeline Parallel.** The main challenge in the Asynchronous Pipeline Parallel (AsyncPP) method (Ajanthan et al., 2025a) is the discrepancy between the gradients at a particular stage and the corresponding weights due to asynchronous updates. Specifically, since the weights of a particular stage are updated multiple times between the forward and backward passes of a microbatch, outdated gradients are used for weight updates. Formally, the weight update for pipe $i$ and stage $j$ can be written as (omitting optimizer specifics):

$$\mathbf{w}_{ij}^{t+1} = \mathbf{w}_{ij}^t - \eta_t \, \nabla f_j(\mathbf{w}_{ij}^{t-\delta_j}; \mathcal{B}_i^{t-\delta_j}) , \qquad (3)$$

where $\eta_t > 0$ is the learning rate, $\mathcal{B}_i^{t-\delta_j}$ is the minibatch, and $\delta_j \geq 0$ is the stage-dependent constant delay due to asynchronous PP updates.

To compensate for this delay, AsyncPP (Ajanthan et al., 2025a) extrapolates the last update direction $(\mathbf{w}_{ij}^t - \mathbf{w}_{ij}^{t-1})$ using the Nesterov Accelerated Gradient (NAG) framework (Nesterov, 1983), and shows that it acts as gradient delay correction in the weight space. This was shown to surpass the synchronous GPipe method on some language modeling tasks, and we employ this approach to optimize each pipe in our experiments.

## 2.3 DATA PARALLELISM

In a typical Data Parallelism (DP) setup (Goyal, 2017; Li et al., 2020), each worker computes the gradient of the full model on a minibatch and communicates it to the central parameter server. The server averages the gradients from all workers, performs an optimization step of the server model, and distributes the updated parameters to the workers for the next iteration. This is equivalent to performing gradient based optimization using a larger minibatch, and the consensus constraint in Eq. (2) is maintained.

---

[2]Asynchronous PP methods (Ajanthan et al., 2025a; Narayanan et al., 2021a) assume comparable compute and communication times per stage, but if the inter-stage bandwidth is low, compression (Ramasinghe et al., 2025) is needed to fully eliminate PP communication overhead.

We consider a serverless scenario, which better reflects a decentralized training setup (Ryabinin et al., 2023). In this, each worker performs a local update using its optimizer and averages the updated weights with others (McMahan et al., 2017), or equivalently averages gradients before applying local updates (Ryabinin et al., 2021). Synchronizing weights (instead of gradients) is becoming popular as they can be communicated infrequently to reduce data transfer (Douillard et al., 2025; 2023; Reddi et al., 2020). To this end, we consider the setup where only a small subset is averaged periodically (Beton et al., 2025; Fournier et al., 2024; Lee et al., 2023) which is shown to perform similarly to traditional DP.

**Sparse Parameter Averaging.** In Sparse Parameter Averaging (SPARTA) (Beton et al., 2025; Fournier et al., 2024), after each local update, a small subset of parameters (*e.g.*, 5%) is averaged across workers, reducing data transfer. Adopting this to our 2D mesh is straightforward, as the local update is performed on each pipe, and the sparse averaging is done between stage replicas with a randomly sampled subset. Formally, sparse averaging for stage $j$ with subset $\mathcal{S}_j^t \subset \{1, \ldots, d_j\}$ can be written as (omitting optimizer specifics):

$$\hat{\mathbf{w}}_{ij}^t = \mathbf{w}_{ij}^{t-1} - \eta_{t-1} \nabla f_j(\mathbf{w}_{ij}^{t-1}; \mathcal{B}_i^{t-1}) , \qquad \forall i , \qquad \text{local update ,} \qquad (4)$$

$$w_{ij:\mu}^t = \begin{cases} \frac{1}{m} \sum_i \hat{w}_{ij:\mu}^t , & \text{if } \mu \in \mathcal{S}_j^t , \\ \hat{w}_{ij:\mu}^t , & \text{if } \mu \notin \mathcal{S}_j^t , \end{cases} \qquad \forall i, \mu , \qquad \text{sparse averaging .}$$

Here, $\eta_t > 0$ is the learning rate, $\mathcal{B}_i^{t-1}$ is the minibatch, and $w_{ij:\mu}^t$ is the $\mu$-th element of vector $\mathbf{w}_{ij}^t$.

## 3 Our Method: Optimization in AsyncMesh

We consider AsyncMesh, where both PP and DP axes in the consensus optimization problem (Eq. (2)) are optimized asynchronously. By making the mesh fully asynchronous, our setup ensures *full pipeline utilization* throughout training without interruption, while encouraging consensus among model replicas. Our setup is clearly illustrated in Fig. 1. For optimizing each pipe, we employ the AsyncPP method (refer to Sec. 2.2) that uses a variant of NAG for delay correction within a pipe. For DP, we introduce an asynchronous version of sparse parameter averaging that eliminates the communication overhead due to DP.

### 3.1 Asynchronous Sparse Parameter Averaging

Let us consider a particular stage, and drop the stage index for simplified notation. In asynchronous DP, the local updates in each worker (*i.e.*, PP optimization) do not wait for the averaging operation (*i.e.*, DP communication) to complete. Therefore, the weights at each stage are updated multiple times while the averaging operation is performed between the stage replicas. Thus, the averaged weights are older than the weights at each worker, which leads to *staleness*. Let $\tau$ be the corresponding delay, then, the delayed sparse averaging can be written as:

$$w_{i:\mu}^t = \begin{cases} \bar{w}_{i:\mu}^{t-\tau} , & \text{if } \mu \in \mathcal{S}^{t-\tau} , \\ \hat{w}_{i:\mu}^t , & \text{if } \mu \notin \mathcal{S}^{t-\tau} , \end{cases} \qquad \forall i, \mu , \qquad \text{where } \bar{\mathbf{w}}^t = \frac{1}{m} \sum_i \hat{\mathbf{w}}_i^t . \qquad (5)$$

Compared to Eq. (4), the only difference is that elements in the subset $\mathcal{S}^{t-\tau}$ are set to the *old average* $\bar{w}_{i:\mu}^{t-\tau}$ instead of the new one at time $t$. This delay is detrimental to training as 1) it ignores the local updates between $t-\tau$ and $t$ for the subset, and 2) the weight vector has a discrepancy as some weights correspond to time $t-\tau$ and the rest at time $t$. Therefore, it is essential to compensate for this delay.

### 3.2 Delay Correction via Estimating Average Staleness

Our idea is to approximate the new average $\bar{\mathbf{w}}^t$ using the old average $\bar{\mathbf{w}}^{t-\tau}$ and the estimated *average staleness*. Specifically, we estimate the average staleness in each stage independently using Exponential Moving Average (EMA) of staleness (*i.e.*, weight differences) throughout training. Precisely, we estimate the new average as,

$$\mathbf{d}_i^t \llbracket \mathcal{S}^{t-\tau} \rrbracket = (1 - \lambda_t) \, \mathbf{d}_i^{t-1} \llbracket \mathcal{S}^{t-\tau} \rrbracket + \lambda_t \left( \hat{\mathbf{w}}_i^t \llbracket \mathcal{S}^{t-\tau} \rrbracket - \hat{\mathbf{w}}_i^{t-\tau} \llbracket \mathcal{S}^{t-\tau} \rrbracket \right) , \qquad \text{EMA of staleness ,} \tag{6}$$

$$\tilde{\mathbf{w}}_i^t \llbracket \mathcal{S}^{t-\tau} \rrbracket = \bar{\mathbf{w}}^{t-\tau} \llbracket \mathcal{S}^{t-\tau} \rrbracket + \mathbf{d}_i^t \llbracket \mathcal{S}^{t-\tau} \rrbracket , \qquad \text{estimate of average at } t ,$$

where $[\![\cdot]\!]$ denotes the indicator and $\lambda_t \in (0,1)$ is the EMA coefficient. Then, the delay corrected asynchronous sparse averaging takes the following form:

$$w_{i:\mu}^t = \begin{cases} \tilde{w}_{i:\mu}^t\,, & \text{if } \mu \in \mathcal{S}^{t-\tau}\,, \\ \hat{w}_{i:\mu}^t\,, & \text{if } \mu \notin \mathcal{S}^{t-\tau}\,, \end{cases} \qquad \forall\, i,\mu\,. \tag{7}$$

Intuitively, the idea here is that $\mathbf{d}_i^t$ being the EMA of staleness, robustly estimates the average staleness. Therefore, $\tilde{\mathbf{w}}_i^t$ approximates the average at time $t$. Concretely,

$$\tilde{\mathbf{w}}_i^t = \bar{\mathbf{w}}^{t-\tau} + \mathbf{d}_i^t \approx^a \bar{\mathbf{w}}^{t-\tau} + \mathbb{E}\big[\hat{\mathbf{w}}_i^t - \hat{\mathbf{w}}_i^{t-\tau}\big] \approx^b \bar{\mathbf{w}}^{t-\tau} + \bar{\mathbf{w}}^t - \bar{\mathbf{w}}^{t-\tau} = \bar{\mathbf{w}}^t\,. \tag{8}$$

Here, the approximation $a$ is due to EMA being a stochastic approximation of the expected value (Robbins & Monro, 1951), and $b$ is due to the empirical average of DP replicas. Note the accuracy of the EMA approximation depends on the delay $\tau$ (lower the better) and the smoothness of the optimization trajectory.[3] Whereas the approximation error of $b$ reduces with increasing number of DP replicas.

Note that sparse averaging is performed after every local step in our description so far. However, this requirement can be relaxed and averaging can be performed every $K$ steps, similar to (Douillard et al., 2023; McMahan et al., 2017). In this setup, our approach is a strict generalization of the concurrent idea of eager DiLoCo (Kale et al., 2025), in which, all the parameters are communicated, the delay $\tau = K$, and $\mathbf{d}_i^t = \frac{1}{m}\big(\hat{\mathbf{w}}_i^t - \hat{\mathbf{w}}_i^{t-\tau}\big)$.

### 3.3 THEORETICAL INSIGHTS

Recall that our aim is to optimize the consensus objective Eq. (2) with asynchronous updates for both PP and DP. AsyncPP (Ajanthan et al., 2025a) proved convergence with fixed delay, providing a theoretical justification for the single pipeline setup. However, it is unclear if sparse averaging ensures consensus, and since we also make it asynchronous, the effect of staleness in achieving consensus needs to be studied.

To this end, we take a first step in theoretically understanding the conditions required for achieving consensus for asynchronous sparse averaging. First, we show that sparse averaging ensures consensus on expectation if the learning rate is chosen proportional to the subset size. Then, we provide a theoretical insight showing that under standard assumptions of stochastic approximation (Robbins & Monro, 1951) with an identical setup for each replica, the EMA approximates the average staleness and consequently ensures consensus on expectation. Both of these provide a theoretical justification that our approach ensures consensus, and when coupled with the standard convergence proof for Stochastic Gradient Descent (SGD) (Bottou et al., 2018), show that our approach converges to a fixed point of Eq. (2).

Suppose the consensus error is defined as:

$$\big\|\boldsymbol{\Delta}^t\big\|^2 \coloneqq \sum_{i=1}^m \big\|\mathbf{w}_i^t - \mathbf{w}^t\big\|^2\,, \qquad \text{where } \mathbf{w}^t = \tfrac{1}{m}\mathbf{w}_i^t\,. \tag{9}$$

We first show that, on expectation, the consensus error vanishes, *i.e.*, all model replicas converge to their average. Now, by invoking the standard convergence proof for SGD (Bottou et al., 2018) on the averaged model, one can show convergence for sparse parameter averaging.

**Theorem 1 (Sparse averaging ensures consensus).** *Let $f$ be a $L$-smooth function, the stochastic gradient be an unbiased estimate of $\nabla f$ and have bounded variance, and $p > 0$ be the averaging probability for an element $\mu \in \{1, \ldots, d\}$, then, for an appropriate choice of learning rate $\eta_t > 0$, the consensus error for updates in Eq. (4) diminishes on expectation,* i.e., $\lim_{t\to\infty} \mathbb{E}\big[\big\|\boldsymbol{\Delta}^t\big\|^2\big] = 0$.

*Proof.* We first show that sparse averaging shrinks the consensus error by a factor of $(1-p)$ on expectation. Then, we derive a recursion on $\mathbb{E}\big[\big\|\boldsymbol{\Delta}^t\big\|^2\big]$ and choose a learning rate proportional to $p$ to ensure that it is a contraction. Detailed proof is provided in the appendix. $\qquad\square$

---

[3]Since EMA is estimated across different time steps, a smoother trajectory (*i.e.*, slow drift in average staleness) leads to a better approximation, conditions for this are stated in the next section.

Consensus of partial averaging has been previously studied in federated learning for pre-determined subsets (Lee et al., 2023), where a similar relationship between the learning rate and subset size is observed. This suggests convergence may be slow for small subset sizes (equivalently, small $p$) due to small learning rates. However, the tightness of this result is unclear, and we leave any such analysis for future work. In practice, we use the standard learning rate value and do *not* adjust it based on the subset size.

For the theoretical analysis of asynchronous sparse averaging, we consider a *homogeneous setup* with the same initialization, optimizer hyperparameters, and i.i.d. data subsets, in which we can show that the expected staleness can be independently estimated in each replica. Suppose the expected staleness and its drift be:

$$\mathbf{D}^t := \mathbb{E}\big[\hat{\mathbf{w}}_i^t - \hat{\mathbf{w}}_i^{t-\tau}\big] , \qquad \boldsymbol{\alpha}_t := \mathbf{D}^t - \mathbf{D}^{t-1} . \tag{10}$$

With standard assumptions from stochastic approximation theory (Robbins & Monro, 1951; Robbins & Siegmund, 1971) such as diminishing EMA coefficient, and diminishing drift in expected staleness, we show that asynchronous (*i.e.*, delayed) averaging ensures consensus on expectation. This, together with Theorem 1, provides a theoretical justification for asynchronous sparse averaging.

**Theorem 2 (Delayed averaging with EMA ensures consensus).** *Consider a homogeneous setup where the average staleness $\mathbf{D}^t$ is bounded and its drift is diminishing,* i.e., $\lim_{t\to\infty} \|\boldsymbol{\alpha}_t\| = 0$. *Then, the EMA based delay correction as in Eq. (6) with $\lambda_t$ satisfying $\sum_t \lambda_t = \infty$, $\sum_t \lambda_t^2 < \infty$, and $\sum_t \frac{\|\boldsymbol{\alpha}_t\|^2}{\lambda_t} < \infty$ ensures consensus on expectation,* i.e., $\lim_{t\to\infty} \mathbb{E}\Big[\big\|\boldsymbol{\Delta}^t\big\|^2\Big] = 0$.

*Proof.* Intuitively, in a homogeneous setup, a particular weight trajectory is equally probable for all replicas, and hence, the expected staleness can be independently estimated in each replica. Moreover, classical stochastic approximation theory (Robbins & Monro, 1951) shows that EMA approximates the expected value. Here, since $\mathbf{D}^t$ is time varying, we use the additional assumption on the interplay between the drift and the EMA coefficient to bound the consensus error. Detailed proof is provided in the appendix. □

The assumption of diminishing $\|\boldsymbol{\alpha}_t\|$ and $\sum_t \frac{\|\boldsymbol{\alpha}_t\|^2}{\lambda_t} < \infty$ can be interpreted as the optimization trajectory being smoother such that the average staleness changes slowly with time and its difference diminishes towards convergence. This also puts a restriction on the allowed delay $\tau$. To this end, sparse averaging with asynchronous updates is appealing as we can reduce the delay $\tau$ by only communicating a small subset at each iteration over a bandwidth limited interconnect.

Note the theory recommends diminishing EMA coefficient $\lambda_t$. In our implementation, we initialize $\lambda_t$ to $0.5$ and after 1k iterations, we gradually decay it to $0.01$ using a cosine schedule. This slightly improves over a constant $\lambda_t$, and we keep this schedule fixed for all the experiments.

## 4 RELATED WORK

**Communication Efficient DP Methods.** DP is a traditional distributed training setup (Goyal, 2017; Li et al., 2020), where each device computes the gradients of the full model in its data split, aggregates the gradients on a central server, performs a central optimization step, and distributes the updated model parameters back to the devices for the next iteration. To reduce the communication overhead, the gradients can be compressed using low-rank approximation (Vogels et al., 2019; Zhao et al., 2024), sparsification (Wang et al., 2017; Wangni et al., 2018), and quantization (Bernstein et al., 2018; Wang et al., 2023). Alternatively, in the serverless setup, each device individually updates the model and periodically synchronizes the model parameters, where partial averaging (Beton et al., 2025; Fournier et al., 2024; Lee et al., 2023) and/or infrequent communication as in DiLoCo (Douillard et al., 2025; 2023; Reddi et al., 2020) reduce communication overhead.

**Asynchronous DP Methods.** Unlike synchronous methods, asynchronous DP methods can fully eliminate the communication overhead. These are well-studied for the parameter server setup that communicates the gradients, and many gradient delay correction mechanisms have been developed (Agarwal & Duchi, 2011; Assran et al., 2020; Stich & Karimireddy, 2019). Notable methods that improve over the simple asynchronous SGD (Recht et al., 2011) include delay dependent learning rate (Barkai et al., 2019; Mishchenko et al., 2022), gradient forecasting with second-order information (Zheng et al., 2017), and look-ahead in the weight space (Hakimi et al., 2019). Apart from this, training dynamics of such asynchronous DP methods have also been analyzed (Mitliagkas

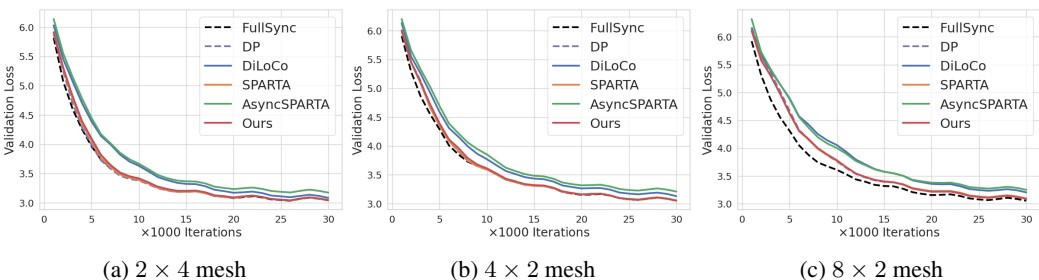

|  (a) $2 \times 4$ mesh | (b) $4 \times 2$ mesh | (c) $8 \times 2$ mesh |

Figure 2: *Results on WikiText with varying mesh configurations with AsyncPP for all methods except FullSync. In all scenarios, our method matches the performance of the fully synchronous method, while outperforming the fully asynchronous baseline AsyncSPARTA.*

et al., 2016). These methods are not directly applicable in our setup, as we communicate weights instead of gradients between replicas. Asynchronous methods are underexplored in this context, although there are a few recent empirical methods designed for DiLoCo (Ajanthan et al., 2025b; Kale et al., 2025; Liu et al., 2024b), we show some of them (Kale et al., 2025) can be viewed as special cases of our method.

**Asynchronous PP Methods.** Asynchronous PP methods eliminate the synchronization bottleneck in PP (Guan et al., 2024) to achieve 100% pipeline utilization at the cost of gradient staleness. In PP, weight stashing is used to ensure correct backpropagation is performed (Narayanan et al., 2019; 2021a), however, the discrepancy (*i.e.*, staleness) between weights and gradients still needs to be compensated. For this, many correction mechanisms such as learning rate discounting (Yang et al., 2021), direct weight prediction (Chen et al., 2018; Guan et al., 2019), and extrapolation in the weight space (Ajanthan et al., 2025a; Zuo et al., 2025) have been developed.

In this work, for the first time, we consider AsyncMesh, where both DP and PP are asynchronous. For PP, we adopt the recent weight extrapolation method (Ajanthan et al., 2025a), and for DP, we combine sparse averaging (Beton et al., 2025) with asynchronous updates to fully eliminate the DP communication overhead.

## 5 EXPERIMENTS

### 5.1 EXPERIMENTAL SETUP

We evaluate on four large-scale language modelling datasets, namely, WikiText (WT) (Merity et al., 2016), BookCorpus (BC) (Zhu et al., 2015), OpenWebText (OWT) (Gokaslan et al., 2019), and FineWeb (FW) (Penedo et al., 2024), using decoder-only architectures with varying mesh configurations, denoted by `PP-stages × DP-replicas`. Our architecture is based on NanoGPT (Karpathy, 2022) with no dropout. The base model has a context length of 1024, an embedding dimension of 768, 12 attention heads, and 12 layers, with approximately 163M parameters. We use the GPT2 tokenizer (Radford et al., 2019) and train the model from scratch. For AsyncPP, NAdamW optimizer (Dozat, 2016) with momentum 0.99 is used as per (Ajanthan et al., 2025a), and all other hyperparameters are set to the standard values for this task (refer to appendix), and kept constant for all the experiments.

Our aim is to show that neither asynchronous updates for both PP and DP, nor the sparse averaging for DP, deteriorate the validation performance, in various configurations. For fair comparison, we compute the validation loss (and perplexity) on the model averaged across all DP replicas (*i.e.*, consensus model), for all the methods. Primarily, we compare our method against **FullSync**, which is the ideal synchronized setup *without any communication efficient scheduling* for PP and only utilizes $\frac{1}{P}$% of the pipeline (Huang et al., 2019; Yang et al., 2021). In addition, we evaluate the following synchronous DP methods: `DP` that averages all the parameters at every step, SPARTA (Beton et al., 2025) that averages a small subset, DiLoCo (Douillard et al., 2023) that performs infrequent communication, and AsyncSPARTA that performs asynchronous sparse averaging without delay correction. For sparse averaging methods, the subset size is 5%, the averaging interval is 1, and the asynchronous methods incur a 10-step delay, unless specified otherwise.

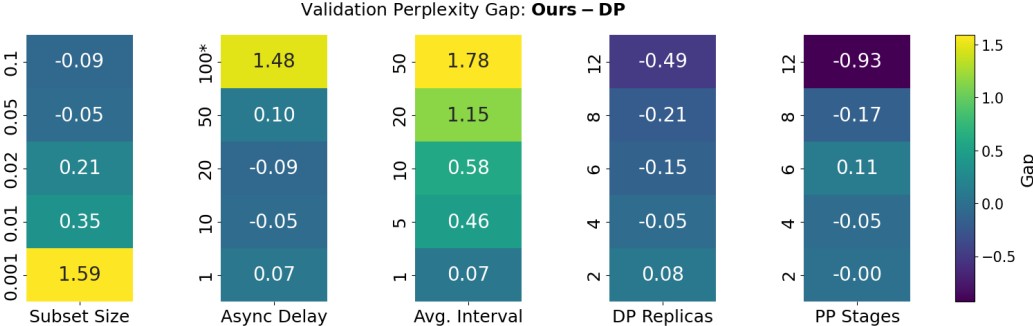

Figure 3: *Perplexity gap (lower the better) between our method and* DP *on WikiText for different configurations. Our method is robust to a range of configurations and scales favourably with the number of DP replicas and PP stages. Here, 100\* indicates,* async-delay = 10 *with* avg-interval = 10, *simulating 100-step effective delay, as* async-delay = 100 *did not converge.*

Our method is implemented in PyTorch (Paszke et al., 2019) using the publicly available codes of SPARTA[4], and AsyncPP[5]. Unless otherwise specified, all experiments use the base architecture described above and are performed on the WikiText dataset. All our experiments are performed on a system equipped with 8 A100 GPUs, and where needed, multiple such instances were used.

## 5.2 MAIN RESULTS

We analyze the validation loss trajectories for different mesh configurations with AsyncPP in Fig. 2. In all scenarios, our method outperforms AsyncSPARTA and matches the performance of FullSync. As shown in the appendix, the behaviour is similar for synchronous PP updates as well. Except AsyncSPARTA and DiLoCo, all methods show near identical performance in most cases. Note DiLoCo is a prominent method in the DP setup with full model in each replica, however, it seems to be inferior in the mesh setup with AsyncPP. To our knowledge, this is the first time DiLoCo is tested with AsyncPP.

In Table 1, we report the validation perplexities on multiple datasets after 30k iterations for the $4 \times 2$ mesh. Our method outperforms AsyncSPARTA by $3 - 6$ points, and yields similar perplexities as FullSync, even surpassing it in 2 out of 4 datasets. This is remarkable as our method is fully asynchronous and only averages 5% of the parameters at each iteration. Note these results are with 2 DP replicas, and as shown in Fig. 3, the more replicas, the better for our method. For completeness, we trained to the compute-optimal point (Hoffmann et al., 2022) on FineWeb, where the perplexities are 19.92 for FullSync, and 20.10 for ours, con-

| Method | WT | BC | OWT | FW |
|---|---|---|---|---|
| FullSync | 21.23 | 35.99 | **35.71** | **36.77** |
| DP | 21.26 | 35.24 | 35.97 | 36.81 |
| SPARTA | 21.30 | 35.15 | 35.73 | 37.10 |
| AsyncSPARTA | 24.80 | 37.83 | 41.41 | 43.20 |
| Ours | **21.14** | **35.09** | 36.13 | 37.31 |

Table 1: *Validation perplexity scores at 30k iterations for the $4 \times 2$ mesh. Our method outperforms AsyncSPARTA, and matches FullSync while eliminating the DP communication overhead. Except FullSync, all other methods use AsyncPP.*

firming the merits of our method beyond doubt. These results demonstrate that our delay correction method effectively compensates for staleness.

## 5.3 VARYING CONFIGURATIONS

To test the robustness of our method in a variety of setups, we consider the base model in the following default setup: subset-size = 5%, async-delay = 10, avg-interval = 1, DP-replicas = 4, PP-stages = 4, and change one criterion at a time. To isolate the effect of our asynchronous sparse averaging, we measure the gap in validation perplexity between our method and DP, which synchronously averages all parameters, while pipeline is optimized with AsyncPP for both methods.

---

[4]https://github.com/matttreed/diloco-sim
[5]https://github.com/PluralisResearch/AsyncPP

As reported in Fig. 3, our method is robust for a wide range of configurations, and it is consistently better than `DP` in many scenarios. Notably, our method is robust for subset size as low as 1% (only a fractional change in perplexity), tolerates delay up to 50 steps, and works reasonably even if we average every 10 steps (*i.e.*, $10\times$ less communication). More encouraging results are that *our method becomes better with a larger mesh* – with a larger number of DP replicas or a larger number of PP stages, the Gap: Ours - `DP` becomes more negative. This shows the superior scalability of our method compared to synchronously averaging all the parameters as in `DP`.

Note that our theory predicts (refer to Sec. 3.2) that a larger number of DP replicas would yield a better approximation of average staleness, leading to better delay correction. These results seem to corroborate that. Meanwhile, AsyncPP is shown to degrade with more PP stages (Ajanthan et al., 2025a), however, the gains from our sparse averaging may help offset this.

**Practical Implications.** Compared to SPARTA, our method achieves $\mathbf{1.5\times - 3.7\times}$ speed-up according to our communication time measurements (refer to appendix for detailed results), with larger meshes yielding better speed-ups. Furthermore, the above results show that our method tolerates a 50-step delay with only a 5% subset size, reducing communication by $10\times$ per iteration (subset and indices need to be exchanged). For simplicity, assuming 1s per forward-backward pass in a stage, this gives 50s for DP communication – enough to support up to 1.5B parameters (FP32) per stage over a 100 Mbps connection, highlighting the viability of decentralized training over the internet.

### 5.4 INCREASING THE MODEL SIZE

To demonstrate the scalability of our approach, we train a *1B parameter model* in the asynchronous 2D mesh. We maintain the number of stages at 4, but increase the embedding dimension to 2304, with 24 attention heads. As illustrated in Fig. 4, the results align with those of the base model. Specifically, our approach matches FullSync throughout training, yielding validation perplexity of **19.53** compared to 19.62 for FullSync. This large-scale experiment demonstrates the merits of our method and the feasibility of asynchronous PP and DP optimization for language model training.

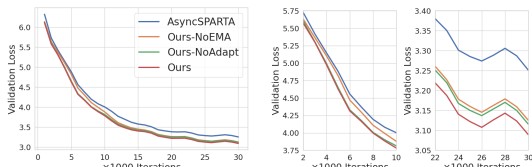

Figure 4: *Results on the 1B parameter model on FineWeb for $4 \times 2$ mesh. Our method matches FullSync, similar to the base model.*

### 5.5 ABLATION STUDY

To understand the effect of different components of our method, we perform an ablation study in $8 \times 2$ mesh on WikiText. Simply using weight differences within each replica, *i.e.*, $\mathbf{d}_i^t = \hat{\mathbf{w}}_i^t - \hat{\mathbf{w}}_i^{t-\tau}$, substantially improve AsyncSPARTA, where EMA and adaptive momentum coefficient as predicted by our theory further improves. Note that the effect of EMA is more prominent in the early phase of training, where the step sizes are larger (*i.e.*, noisier). This aligns with our intuition that EMA robustly approximates the average staleness.

Figure 5: *Ablation of our method on WikiText for the $8 \times 2$ configuration. EMA improves the early phase of training, and the adaptive momentum coefficient yields further marginal improvement.*

## 6 CONCLUSION

In this paper, we studied a new AsyncMesh setup where both DP and PP are asynchronous, and introduced a fully asynchronous approach that can match the performance of the fully synchronized method. We theoretically showed that our method converges to a fixed point of the consensus objective on expectation, despite sparse averaging and asynchronous communication. Our experiments on a wide range of configurations demonstrate the merits of our approach, and show the feasibility of asynchronous optimization for large scale language model training. By alleviating communication overhead without any performance penalty, our approach takes a step towards realizing large-scale collaborative training over the internet.

## REPRODUCIBILITY STATEMENT

All assumptions related to the theoretical claims are clearly mentioned in the theorem statements. Furthermore, necessary details to understand the experiments and results are provided in the main paper, and all hyperparameter settings are provided in the appendix for reproducibility. Our anonymized code is submitted as supplementary material and will be made publicly available upon acceptance.

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

## A   THEORETICAL INSIGHTS

We restate the problem setup, introduce some notations, and then turn to the proofs.

### A.1   SETUP AND NOTATIONS

We consider the DP setup without PP for simplified theoretical analysis. In this, the consensus objective takes the following form:

$$\min_{\mathbf{w} \in \mathbb{R}^d} f(\mathbf{w}; \mathcal{D}) := \min_{\mathbf{w}_i \in \mathbb{R}^d} \sum_{i=1}^{m} f(\mathbf{w}_i; \mathcal{D}_i) , \tag{11}$$

$$\text{s.t.} \quad \mathbf{w}_i = \mathbf{w} , \qquad \forall\, i \in \{1, \dots, m\} .$$

Here, $i \in \{1, \dots, m\}$ denotes the workers, $\mathbf{w}_i, \mathcal{D}_i$ denote the model weights and data chunk for worker $i$, and $f$ is the loss function. Typically, $\mathcal{D}_i$ is an i.i.d. subset of $\mathcal{D}$.

Let us first define some notations:

$$\bar{\mathbf{w}}^t := \frac{1}{m} \sum_{i=1}^{m} \mathbf{w}_i^t , \qquad\qquad \text{averaged weights} , \tag{12}$$

$$\mathbf{g}_i^t := \nabla f_i(\mathbf{w}_i^t; \mathcal{B}_i^t) , \qquad\qquad \text{minbatch gradient} ,$$

$$\bar{\mathbf{g}}^t := \frac{1}{m} \sum_{i=1}^{m} \mathbf{g}_i^t , \qquad\qquad \text{average of gradients} ,$$

$$\left\| \mathbf{\Delta}^t \right\|^2 := \sum_{i=1}^{m} \left\| \mathbf{w}_i^t - \bar{\mathbf{w}}^t \right\|^2 , \qquad\qquad \text{consensus error} .$$

Here, $\left\| \mathbf{\Delta}^t \right\|^2$ denotes the post-averaging consensus error at iteration $t-1$, and the corresponding pre-averaging error is denoted as $\left\| \hat{\mathbf{\Delta}}^t \right\|^2$. Analogously, $\mathbf{\Delta}_i^t := \mathbf{w}_i^t - \bar{\mathbf{w}}^t$.

We are now ready to prove that sparse averaging ensures consensus across the weights of all workers on expectation.

### A.2   SPARSE AVERAGING ENSURES CONSENSUS

**Lemma 1.** *Let $p$ be the probability of an element $\mu \in \mathcal{S}^t$ independent of others, then sparse averaging shrinks the consensus error by a factor of $(1-p)$ on expectation, i.e.,*
$\mathbb{E}\left[ \left\| \mathbf{\Delta}^t \right\|^2 \right] = (1-p)\, \mathbb{E}\left[ \left\| \hat{\mathbf{\Delta}}^t \right\|^2 \right].$

*Proof.* By definition of sparse averaging, $w_{i:\mu}^{t+1} = \frac{1}{m} \sum_i \hat{w}_{i:\mu}^t$ for all $\mu \in \mathcal{S}^t$. Therefore,

$$\mathbb{E}\left[ \left\| \mathbf{\Delta}^t \right\|^2 \right] = \sum_{i=1}^{m} \sum_{\mu=1}^{d} \mathbb{E}\left[ \left( \mathbf{\Delta}_{i:\mu}^t \right)^2 \right] , \qquad \text{linearity of expectation} , \tag{13}$$

$$= \sum_{i=1}^{m} \sum_{\mu=1}^{d} \left( \hat{\mathbf{\Delta}}_{i:\mu}^t \right)^2 \mathbb{E}\left[\!\left[ \mu \notin \mathcal{S}^t \right]\!\right] , \qquad \left[\!\left[ \cdot \right]\!\right] \text{ is the indicator} ,$$

$$= (1-p)\, \mathbb{E}\left[ \left\| \hat{\mathbf{\Delta}}^t \right\|^2 \right] . \qquad P\!\left[\!\left[ \mu \notin \mathcal{S}^t \right]\!\right] = 1 - p .$$

$\square$

**Theorem 3.** *Let $f$ be a $L$-smooth function, the stochastic gradient $\mathbf{g}_i^t$ be an unbiased estimate of $\nabla f$ and have bounded variance $\sigma^2$, and $p > 0$ be the averaging probability for an element $\mu$, then, for an appropriate choice of learning rate $\eta_t > 0$ the consensus error diminishes on expectation, i.e.,*
$\lim_{t \to \infty} \mathbb{E}\left[ \left\| \mathbf{\Delta}^t \right\|^2 \right] = 0.$

*Proof.* Let us expand $\hat{\boldsymbol{\Delta}}_i^{t+1}$:

$$\hat{\boldsymbol{\Delta}}_i^{t+1} = \hat{\mathbf{w}}_i^{t+1} - \frac{1}{m} \sum_i \hat{\mathbf{w}}_i^{t+1} \,, \tag{14}$$

$$= \mathbf{w}_i^t - \eta_t \, \mathbf{g}_i^t - \frac{1}{m} \sum_i \left( \mathbf{w}_i^t - \eta_t \, \mathbf{g}_i^t \right) \,, \qquad \text{local update} \,,$$

$$= \boldsymbol{\Delta}_i^t - \eta_t \left( \mathbf{g}_i^t - \bar{\mathbf{g}}^t \right) \,.$$

Now, the pre-averaging consensus error can be written as:

$$\left\| \hat{\boldsymbol{\Delta}}^{t+1} \right\|^2 = \sum_i \left\| \boldsymbol{\Delta}_i^t - \eta_t \left( \mathbf{g}_i^t - \bar{\mathbf{g}}^t \right) \right\|^2 \,, \tag{15}$$

$$= \sum_i \left( \left\| \boldsymbol{\Delta}_i^t \right\|^2 - 2\,\eta_t\, \boldsymbol{\Delta}_i^t \cdot \left( \mathbf{g}_i^t - \bar{\mathbf{g}}^t \right) + \eta_t^2 \left\| \mathbf{g}_i^t - \bar{\mathbf{g}}^t \right\|^2 \right) \,, \qquad \cdot \text{ is the dot product} \,,$$

$$= \sum_i \left( \left\| \boldsymbol{\Delta}_i^t \right\|^2 - 2\,\eta_t\, \boldsymbol{\Delta}_i^t \cdot \mathbf{g}_i^t + \eta_t^2 \left\| \mathbf{g}_i^t - \bar{\mathbf{g}}^t \right\|^2 \right) \,, \qquad \textstyle\sum_i \boldsymbol{\Delta}_i^t = 0 \,.$$

We now bound each of the terms following techniques similar to that of (Yu et al., 2019). Consider $\left\| \mathbf{g}_i^t - \bar{\mathbf{g}}^t \right\|^2$:

$$\left\| \mathbf{g}_i^t - \bar{\mathbf{g}}^t \right\|^2 = \left\| \mathbf{g}_i^t - \nabla f(\mathbf{w}_i^t) + \nabla f(\mathbf{w}_i^t) - \bar{\nabla} f(\mathbf{w}_i^t) + \bar{\nabla} f(\mathbf{w}_i^t) - \bar{\mathbf{g}}^t \right\|^2 \,, \tag{16}$$

$$\leq 3 \left\| \mathbf{g}_i^t - \nabla f(\mathbf{w}_i^t) \right\|^2 + 3 \left\| \nabla f(\mathbf{w}_i^t) - \bar{\nabla} f(\mathbf{w}_i^t) \right\|^2 + 3 \left\| \bar{\nabla} f(\mathbf{w}_i^t) - \bar{\mathbf{g}}^t \right\|^2 \,,$$

where $\bar{\nabla} f(\mathbf{w}_i^t) = \frac{1}{m} \sum_i \nabla f(\mathbf{w}_i^t)$, and the second step is due to $(a + b + c)^2 \leq 3(a^2 + b^2 + c^2)$. Each of these terms can be bounded as follows: 1) using bounded variance,

$$\sum_i \left\| \mathbf{g}_i^t - \nabla f(\mathbf{w}_i^t) \right\|^2 \leq m\,\sigma^2 \,. \tag{17}$$

2) using $L$-smoothness and triangle inequality,

$$\sum_i \left\| \nabla f(\mathbf{w}_i^t) - \bar{\nabla} f(\mathbf{w}_i^t) \right\|^2 = \sum_i \left\| \nabla f(\mathbf{w}_i^t) - \nabla f(\bar{\mathbf{w}}^t) + \nabla f(\bar{\mathbf{w}}^t) - \bar{\nabla} f(\mathbf{w}_i^t) \right\|^2 \,, \tag{18}$$

$$= \sum_i 2 \left\| \nabla f(\mathbf{w}_i^t) - \nabla f(\bar{\mathbf{w}}^t) \right\|^2 + 2 \left\| \frac{1}{m} \sum_i \nabla f(\bar{\mathbf{w}}^t) - \frac{1}{m} \sum_i \nabla f(\mathbf{w}_i^t) \right\|^2 \,,$$

$$\leq 2L^2 \sum_i \left\| \mathbf{w}_i^t - \bar{\mathbf{w}}^t \right\|^2 + \frac{2L^2}{m} \sum_k \sum_i \left\| \mathbf{w}_i^t - \bar{\mathbf{w}}^t \right\|^2 \,,$$

$$= 4L^2 \left\| \boldsymbol{\Delta}^t \right\|^2 \,.$$

3) using triangle inequality and bounded variance,

$$\sum_i \left\| \bar{\nabla} f(\mathbf{w}_i^t) - \bar{\mathbf{g}}^t \right\|^2 \leq \frac{1}{m} \sum_i \left\| \mathbf{g}_i^t - \nabla f(\mathbf{w}_i^t) \right\|^2 \leq \sigma^2 \,. \tag{19}$$

Altogether for some constant $C > 0$, we can write:

$$\mathbb{E}\left[ \sum_i \left\| \mathbf{g}_i^t - \bar{\mathbf{g}}^t \right\|^2 \right] \leq C \left( m\,\sigma^2 + L^2\, \mathbb{E}\left[ \left\| \boldsymbol{\Delta}^t \right\|^2 \right] \right) \,. \tag{20}$$

Now, consider the term $\mathbb{E}[\sum_i \boldsymbol{\Delta}_i^t \cdot \mathbf{g}_i^t]$:

$$\mathbb{E}\left[\sum_i \boldsymbol{\Delta}_i^t \cdot \mathbf{g}_i^t\right] = \mathbb{E}\left[\sum_i \boldsymbol{\Delta}_i^t \cdot \nabla f(\mathbf{w}_i^t)\right] , \qquad \mathbb{E}[\mathbf{g}_i^t \mid \boldsymbol{\Delta}_i^t] = \mathbb{E}[\mathbf{g}_i^t \mid \mathbf{w}_i^t] = \nabla f(\mathbf{w}_i^t) ,$$

$$\tag{21}$$

$$= \mathbb{E}\left[\sum_i \boldsymbol{\Delta}_i^t \cdot \left(\nabla f(\mathbf{w}_i^t) - \nabla f(\bar{\mathbf{w}}^t)\right)\right] , \qquad \sum_i \boldsymbol{\Delta}_i^t = 0 ,$$

$$\leq L\,\mathbb{E}\left[\sum_i \left\|\boldsymbol{\Delta}_i^t\right\| \cdot \left\|\boldsymbol{\Delta}_i^t\right\|\right] , \qquad L\text{-smooth},\ \boldsymbol{\Delta}_i^t = \mathbf{w}_i^t - \bar{\mathbf{w}}^t ,$$

$$= L\,\mathbb{E}\left[\left\|\boldsymbol{\Delta}^t\right\|^2\right] .$$

Putting everything together,

$$\left\|\hat{\boldsymbol{\Delta}}^{t+1}\right\|^2 \leq (1 + 2\,\eta_t\,L)\,\mathbb{E}\left[\left\|\boldsymbol{\Delta}^t\right\|^2\right] + \eta_t^2\,C\left(m\,\sigma^2 + L^2\,\mathbb{E}\left[\left\|\boldsymbol{\Delta}^t\right\|^2\right]\right) . \tag{22}$$

From Lemma 1,

$$\mathbb{E}\left[\left\|\boldsymbol{\Delta}^{t+1}\right\|^2\right] = (1 - p)\,\mathbb{E}\left[\left\|\hat{\boldsymbol{\Delta}}^{t+1}\right\|^2\right] , \tag{23}$$

$$\leq (1 - p)\,(1 + 2\,\eta_t\,L)\,\mathbb{E}\left[\left\|\boldsymbol{\Delta}^t\right\|^2\right] + \mathcal{O}(\eta_t^2) .$$

Note, $\eta_t$ can be chosen such that the quadratic term is vanishes, *i.e.*, $\eta_t > 0, \sum_t \eta_t = \infty,$ and $\sum_t \eta_t^2 < \infty$ (Robbins & Monro, 1951), and the coefficient $(1-p)\,(1 + 2\,\eta_t\,L)$ is strictly less than 1, *i.e.*, $\eta_t < \frac{p}{2(1-p)L}$.

This yields a contraction and ensures $\lim_{t\to\infty} \mathbb{E}\left[\left\|\boldsymbol{\Delta}^t\right\|^2\right] = 0.$ $\qquad\square$

This proves that sparse averaging can lead to consensus among workers, in the sense that, on expectation, all weight vectors converge to their average. This, together with the standard convergence proof of SGD (Bottou et al., 2018) guarantees that sparse averaging converges to a fixed point of Eq. (11).

### A.3 EMA BASED DELAY CORRECTION ENSURES CONSENSUS

We consider a *homogeneous setup* where all workers are initialized to the same point, the data chunks are i.i.d., and the optimizer parameters are identical. In this, we first show that the expected staleness $\mathbb{E}\left[\hat{\mathbf{w}}_i^t - \hat{\mathbf{w}}_i^{t-\tau}\right]$ can be independently estimated in the worker $i$. Then, we prove that the drift of $\mathbf{d}_i^t$ between different workers diminishes, ensuring consensus. In this section, with a slight abuse of notation, we define $\bar{\mathbf{w}}^t = \frac{1}{m}\sum_i \hat{\mathbf{w}}_i^t$.

**Lemma 2.** *In a homogeneous setup as defined above, the expected value of the weight drift is independent of the worker;* i.e., $\mathbb{E}\left[\hat{\mathbf{w}}_i^t - \hat{\mathbf{w}}_i^{t-\tau}\right] = \mathbf{D}^t$.

*Proof.* Considering a single local update:

$$\mathbb{E}\left[\hat{\mathbf{w}}_i^{k+1} - \mathbf{w}_i^k\right] = -\mathbb{E}\left[\eta_k\,\mathbf{g}_i^k\right] = -\eta_k \nabla f(\mathbf{w}_i^k) , \qquad \mathbf{g}_i^k \text{ is unbiased .} \tag{24}$$

This is independent and identical for each worker if $\mathbf{w}_i^k = \mathbf{w}^k$ for all $i$. This argument can be extended to multiple steps for a homogeneous setup due to the same initialization, i.i.d. data samples, and identical optimizer hyperparameters. Intuitively, one can see that a particular weight trajectory is equally probable for all workers in a homogeneous setup.

Additionally, the EMA update parameters ($\mathbf{d}_i^0$ and $\lambda_t$) are also identical across workers. Therefore, the average weight drift $\mathbb{E}\left[\hat{\mathbf{w}}_i^t - \hat{\mathbf{w}}_i^{t-\tau}\right] = \mathbb{E}\left[\hat{\mathbf{w}}_i^t - \mathbf{w}_i^{t-\tau}\right] + \mathbb{E}\left[\mathbf{w}_i^{t-\tau} - \hat{\mathbf{w}}_i^{t-\tau}\right]$ is independent of the worker $i$. $\qquad\square$

**Theorem 4.** *Consider a homogeneous setup where the average staleness $\mathbf{D}^t$ is bounded and its drift $\boldsymbol{\alpha}_t := \mathbf{D}^t - \mathbf{D}^{t-1}$ is diminishing, i.e., $\lim_{t\to\infty} \|\boldsymbol{\alpha}_t\| = 0$. Then, the EMA based delay correction with $\lambda_t$ satisfying $\sum_t \lambda_t = \infty$, $\sum_t \lambda_t^2 < \infty$, and $\sum_t \frac{\|\boldsymbol{\alpha}_t\|^2}{\lambda_t} < \infty$ ensures consensus, i.e., $\lim_{t\to\infty} \mathbb{E}\left[\|\boldsymbol{\Delta}^t\|^2\right] = 0$.*

*Proof.* Now consider $\boldsymbol{\Delta}_i^t$:

$$\boldsymbol{\Delta}_i^t = \mathbf{w}_i^t - \bar{\mathbf{w}}_i^t, \tag{25}$$

$$= \bar{\mathbf{w}}^{t-\tau} + \mathbf{d}_i^t - \frac{1}{m}\sum_i \left(\bar{\mathbf{w}}^{t-\tau} + \mathbf{d}_i^t\right), \qquad\qquad \text{EMA},$$

$$= \mathbf{d}_i^t - \bar{\mathbf{d}}_i^t.$$

Note that, $\bar{\mathbf{d}}_i^t = \frac{1}{m}\sum_i \mathbf{d}_i^t$ is the empirical estimate of $\mathbb{E}[\mathbf{d}_i^t]$. Following the arguments from Lemma 2, we can see that $\mathbb{E}[\mathbf{d}_i^t] = \mathbf{D}^t$ for all workers. If $\mathbf{D}^t$ is a constant, then the standard stochastic approximation theory (Robbins & Monro, 1951) yields the desired result. However, since $\mathbf{D}^t$ varies with time, we need an additional assumption on the interplay between the drift and the EMA coefficient, that $\frac{\|\boldsymbol{\alpha}_t\|^2}{\lambda_t} \to 0$ and carefully bound the consensus error.

Substituting $\bar{\mathbf{d}}_i^t = \mathbf{D}^t$, together with the EMA update:

$$\boldsymbol{\Delta}_i^t = \mathbf{d}_i^t - \mathbf{D}^t, \tag{26}$$

$$= (1 - \lambda_t)\,\mathbf{d}_i^{t-1} + \lambda_t\left(\hat{\mathbf{w}}_i^t - \hat{\mathbf{w}}_i^{t-\tau}\right) - \mathbf{D}^t,$$

$$= (1 - \lambda_t)\left(\mathbf{d}_i^{t-1} - \mathbf{D}^{t-1} + \mathbf{D}^{t-1} - \mathbf{D}^t\right) + \lambda_t\left(\hat{\mathbf{w}}_i^t - \hat{\mathbf{w}}_i^{t-\tau} - \mathbf{D}^t\right),$$

$$= (1 - \lambda_t)\,\boldsymbol{\Delta}_i^{t-1} - (1 - \lambda_t)\,\boldsymbol{\alpha}_t + \lambda_t\,\boldsymbol{\xi}_i^t, \qquad \boldsymbol{\xi}_i^t = \hat{\mathbf{w}}_i^t - \hat{\mathbf{w}}_i^{t-\tau} - \mathbf{D}^t.$$

Now square both sides:

$$\left\|\boldsymbol{\Delta}_i^t\right\|^2 = (1 - \lambda_t)^2\left\|\boldsymbol{\Delta}_i^{t-1}\right\|^2 + (1 - \lambda_t)^2\left\|\boldsymbol{\alpha}_t\right\|^2 + \lambda_t^2\left\|\boldsymbol{\xi}_i^t\right\|^2 \tag{27}$$

$$+ 2(1 - \lambda_t)\lambda_t\,\boldsymbol{\Delta}_i^{t-1}\cdot\boldsymbol{\xi}_i^t - 2(1 - \lambda_t)^2\boldsymbol{\Delta}_i^{t-1}\cdot\boldsymbol{\alpha}_t - 2(1 - \lambda_t)\lambda_t\,\boldsymbol{\alpha}_t\cdot\boldsymbol{\xi}_i^t,$$

where $\cdot$ is the dot-product. Consider the first cross term:

$$\mathbb{E}\left[\boldsymbol{\Delta}_i^{t-1}\cdot\boldsymbol{\xi}_i^t\right] = \mathbb{E}\left[\boldsymbol{\Delta}_i^{t-1}\cdot\mathbb{E}\left[\boldsymbol{\xi}_i^t\right]\right] = 0, \qquad \mathbb{E}[\boldsymbol{\xi}_i^t] = 0 \text{ due to Lemma 2}. \tag{28}$$

Using Young's inequality for the other cross terms, *i.e.*, $2ab \leq \epsilon a^2 + \frac{1}{\epsilon}b^2$ for any $\epsilon > 0$, we can write:

$$\mathbb{E}\left[\left\|\boldsymbol{\Delta}_i^t\right\|^2\right] \leq (1 - \lambda_t)^2\,\mathbb{E}\left[\left\|\boldsymbol{\Delta}_i^{t-1}\right\|^2\right] + (1 - \lambda_t)^2\left\|\boldsymbol{\alpha}_t\right\|^2 + \lambda_t^2\,\mathbb{E}\left[\left\|\boldsymbol{\xi}_i^t\right\|^2\right] \tag{29}$$

$$+ (1 - \lambda_t)^2\left(\epsilon\,\mathbb{E}\left[\left\|\boldsymbol{\Delta}_i^{t-1}\right\|^2\right] + \frac{\|\boldsymbol{\alpha}_t\|^2}{\epsilon}\right) + (1 - \lambda_t)\lambda_t\left(\epsilon\,\mathbb{E}\left[\left\|\boldsymbol{\xi}_i^t\right\|^2\right] + \frac{\|\boldsymbol{\alpha}_t\|^2}{\epsilon}\right).$$

The term $\mathbb{E}\left[\left\|\boldsymbol{\xi}_i^t\right\|^2\right]$ can be bounded due to bounded delay $\tau$. By setting $\epsilon = \lambda_t$, the above can be simplified as:

$$\mathbb{E}\left[\left\|\boldsymbol{\Delta}_i^t\right\|^2\right] \leq (1 - \lambda_t)^2(1 + \lambda_t)\,\mathbb{E}\left[\left\|\boldsymbol{\Delta}_i^{t-1}\right\|^2\right] + \frac{A}{\lambda_t}\left\|\boldsymbol{\alpha}_t\right\|^2 \tag{30}$$

$$+ B\left(\|\boldsymbol{\alpha}_t\|^2 + \lambda_t^2\right), \qquad \text{for some constants } A, B > 0,$$

$$\leq (1 - \lambda_t)\,\mathbb{E}\left[\left\|\boldsymbol{\Delta}_i^{t-1}\right\|^2\right] + \frac{A}{\lambda_t}\left\|\boldsymbol{\alpha}_t\right\|^2 + B\left(\|\boldsymbol{\alpha}_t\|^2 + \lambda_t^2\right), \qquad 0 < \lambda_t < 1.$$

Since $\frac{\|\boldsymbol{\alpha}_t\|^2}{\lambda_t}$, $\|\boldsymbol{\alpha}_t\|^2$, and $\lambda_t^2$ are diminishing, the above can be shown to be a contraction and therefore, $\mathbb{E}\left[\|\boldsymbol{\Delta}_i^t\|^2\right] \to 0$ following the classical stochastic approximation theory (Robbins & Monro, 1951; Robbins & Siegmund, 1971). Consequently, the consensus error vanishes:

$$\mathbb{E}\left[\|\boldsymbol{\Delta}^t\|^2\right] = \mathbb{E}\left[\sum_i\|\boldsymbol{\Delta}_i^t\|^2\right] = \sum_i\mathbb{E}\left[\|\boldsymbol{\Delta}_i^t\|^2\right] \to 0. \tag{31}$$

$\square$

Note that the EMA update and the arguments in the theorem are elementwise, hence, they naturally extend to the PP with sparse averaging setup. This, together with Theorem 3 and the convergence proof of SGD, provides a theoretical justification for convergence for our method in the PP setup with sparse averaging on expectation, despite a fixed delay.

# B  EXPERIMENTS

## B.1  EXPERIMENTAL SETUP

We evaluate on four large-scale language modeling datasets, namely, WikiText (WT) (Merity et al., 2016), BookCorpus (BC) (Zhu et al., 2015), OpenWebText (OWT) (Gokaslan et al., 2019), and FineWeb (FW) (Penedo et al., 2024). For WikiText, we utilize the predefined training and validation splits; for BookCorpus and OpenWebText, we randomly select $10\%$ of the training set as the held-out validation set; and for FineWeb, we use the streaming feature in Huggingface datasets and hold out 10k samples in the stream for validation. Our architecture is based on NanoGPT (Karpathy, 2022) with no dropout. The base model has a context length of 1024, an embedding dimension of 768, 12 attention heads, and 12 layers, with approximately 163M parameters. We use the GPT2 tokenizer (Radford et al., 2019) and train the model from scratch. For configurations with 2 and 4 pipeline stages, equal number of layers are assigned to each stage, unless specified otherwise. For 8 stages, stages $2-5$ are assigned 2 layers, and others have 1 layer each.

Across all experiments, we maintain a microbatch size of 8 per DP replica, a learning rate of $3e\text{-}4$, a weight decay of $0.01$, and gradient clipping norm of 1. For experiments with asynchronous PP, NAdamW optimizer (Dozat, 2016) with momentum $0.99$ is used as per (Ajanthan et al., 2025a). For synchronous PP experiments, GPipe (Huang et al., 2019) with AdamW optimizer (Loshchilov, 2017) is used, and the number of microbatches is set to 2. Each experiment is run for 30k iterations, with a linear warmup of 3k iterations starting from a learning rate of $1e\text{-}7$. Then, it is decayed to $3e\text{-}5$ following a cosine decay schedule. For our method, the EMA variable $\mathbf{d}_i^t$ is initialized to zero.

For DiLoCo, the outer learning rate of 1 performs better (*i.e.*, averaging instead of outer optimization step) in our 2D mesh with AsyncPP, and the outer-update interval is set to 10 steps in our experiments.

**1B Model.**  We maintain the number of stages at 4 with stage assignment of $[1, 3, 4, 4]$ number of layers, but increase the embedding dimension to 2304, with 24 attention heads. The learning rate warmup step is adjusted to 6k for all methods and run for 100k iterations. All other hyperparameters are the same as the base model.

**Varying Configurations.**  When changing a criterion, all other criteria are kept to the default values: `subset-size = 5%`, `async-delay = 10`, `avg-interval = 1`, `DP-replicas = 4`, `PP-stages = 4`. While varying the averaging interval, the asynchronous delay is set to 1, such that the effective delay is equal to the averaging interval.

## B.2  ADDITIONAL RESULTS

We provide additional validation loss trajectories for various methods with synchronous PP in Fig. 6, on multiple datasets corresponding to the results in the main paper in Fig. 7, and for compute optimal training for the base model in Fig. 8. Furthermore, we provide consensus error plots for our method confirming the theory in Fig. 9, validation loss vs time plot for the 1B model in Fig. 10.

**Gain in Wall-clock Time.**  Since we simulate the asynchronous DP setup via buffering, and due to implementation differences between our method and FullSync, their practical wall-clock times are not comparable. However, the theoretical gain in wall-clock time per iteration due to asynchronous sparse averaging is $O(d^2/B)$, where $d$ is the embedding dimension, and $B$ is the bandwidth, as we fully eliminate DP overhead. Since the staleness is in update steps, the ratio between allowed time and the data transfer volume is $O(Bd^3\tau/pd^2) = O(Bd\tau/p)$, where $\tau$ is the allowed delay, $p$ is the subset size, and note the compute time is approximately cubic in $d$ (Ryabinin et al., 2023). Therefore, our method scales favourably for large models. Note, AsyncPP (Ajanthan et al., 2025a) further improves this wall-clock time gain.

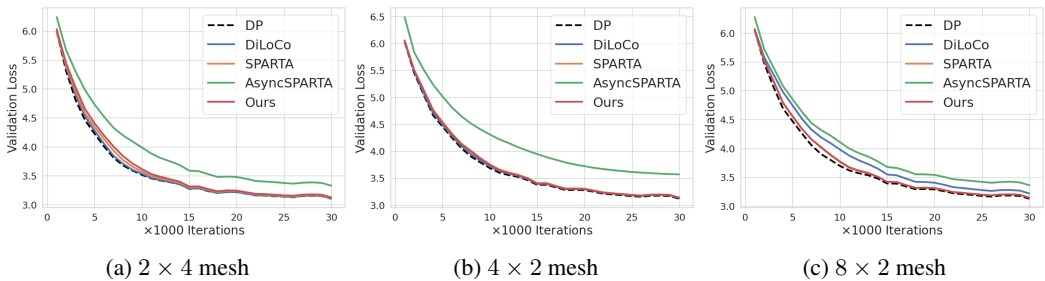

(a) $2 \times 4$ mesh

(b) $4 \times 2$ mesh

(c) $8 \times 2$ mesh

Figure 6: *Results on WikiText with varying mesh configurations with synchronous PP. In all scenarios, our method matches the performance of the fully synchronous method, same as the case with AsyncPP.*

For completeness, we show the empirical time savings that can be achieved by our asynchronous sparse averaging in Table 2. This clearly shows just by making sparse averaging asynchronous with 1-step delay, we can achieve a speed-up of $1.5 \times - 3.7 \times$ depending on the mesh configuration, with larger meshes yielding better speed-ups.

**Heterogeneous Setup** So far, our experiments have been on a homogeneous setup where the compute capability of the devices is the same. To stress test our method, we simulate a heterogeneous setup for a 44 mesh by varying device speeds, namely, H2×: $[2, 1, 2, 3]$, H5×: $[5, 3, 1, 11]$, and H10×: $[10, 8, 1, 21]$, along with the homogeneous setup: H1×: $[1, 1, 1, 1]$, with the provided relative device speeds. To ensure each device initiates the averaging operation (*i.e.*, all-reduce) at approximately the same time, we set the averaging interval proportional to the device speed, *i.e.*, faster devices will perform more iterations between an averaging step. This means, model replicas at different training stages are sparsely averaged with a delay due to asynchronous DP. This is analogous to dynamic local updates in (Liu et al., 2024b). For fair comparison, we fix the total number of iterations across all replicas to be 120k. As shown in Fig. 11, even with up to $10 \times$ difference in device speeds (denoted H10×), and delayed sparse averaging of replicas at different training stages, the degradation of validation loss remains small. This degradation is negligible when accounting for the wall-clock speed-up, which scales with device speeds due to the absence of DP communication overhead.

**Comparison with Other Communication Efficient Methods.** We compare with some existing methods that compress the DP communication via quantization and TopK sample in Fig. 12. The quantization results show sparse averaging methods (even with delay) are more robust to quantization, allowing further reduction in DP communication requirements. Finally, we compare with the concurrent work of eager updates for DiLoCo (Kale et al., 2025) in Fig. 13, showing that our method strictly generalizes it.

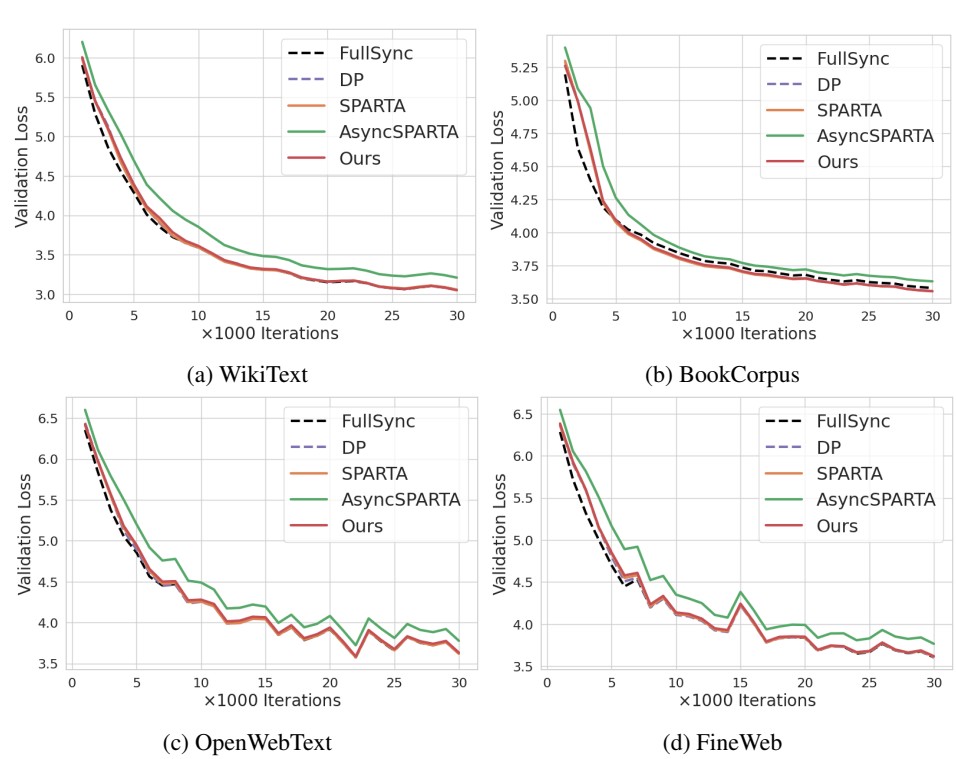

(a) WikiText              (b) BookCorpus

(c) OpenWebText            (d) FineWeb

Figure 7: *Results on different datasets for* $4 \times 2$ *mesh. Our method performs similarly to FullSync in all scenarios, demonstrating virtually no performance degradation due to staleness or sparse averaging.*

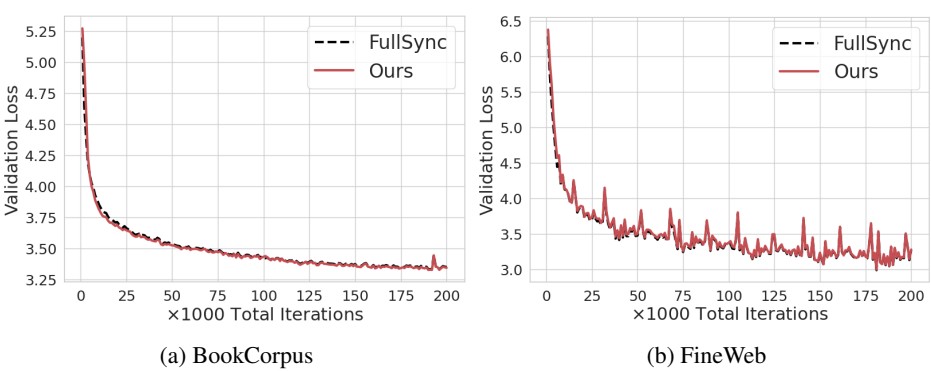

(a) BookCorpus            (b) FineWeb

Figure 8: *Compute optimal training (Hoffmann et al., 2022) for BookCorpus and FineWeb for the base model with* $4 \times 2$ *mesh. Our method is nearly identical to FullSync for longer training as well, validating its merits. The final validation perplexities are, for BookCorpus,* FullSync: 28.02, *and* Ours: 27.86 *and for FineWeb,* FullSync: 19.92, *and* Ours: 20.10. *The validation curve for FineWeb is noisy for both methods, probably due to the way the validation set is selected from the Huggingface stream.*

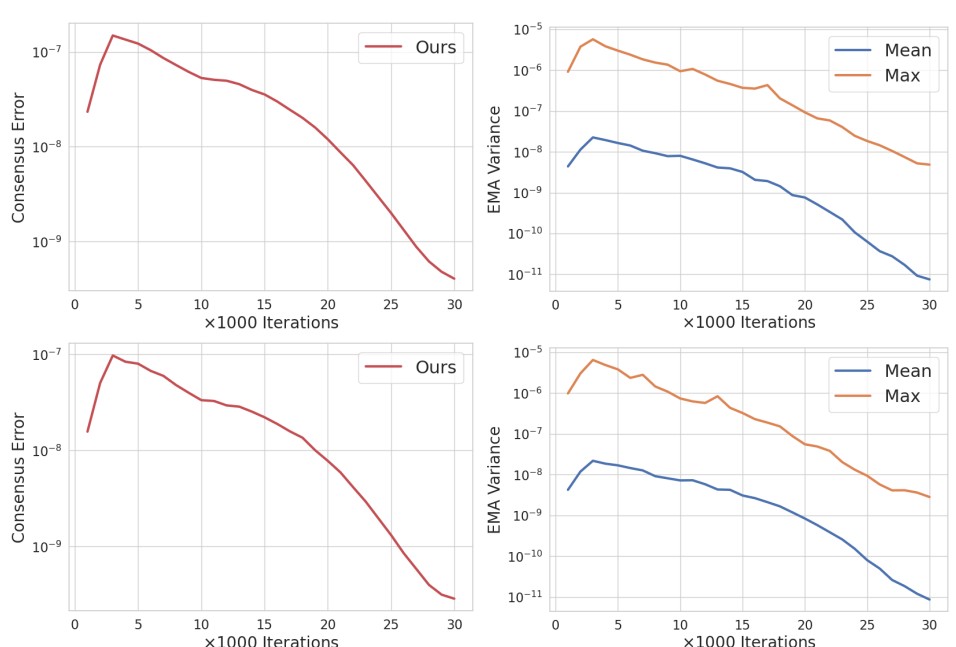

Figure 9: *Mean consensus error $\frac{1}{md}\left\|\boldsymbol{\Delta}^t\right\|^2$ as in Eq. (12) and variance between EMA estimates ($\mathbf{d}_i^t$) in each replica, for our method on the $2 \times 4$ (top) and $4 \times 2$ mesh (bottom). For EMA, the mean and max across the model dimension are shown. Results perfectly align with the theory that independent EMA estimates in each replica converge to the expected value (*i.e., variance vanishes), and the consensus error for our method diminishes to zero.*

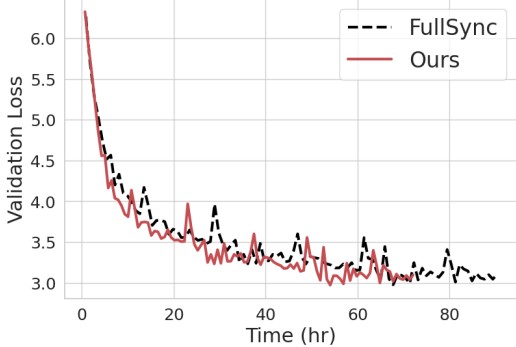

Figure 10: *Validation loss vs time for 1B model. Even with suboptimal implementation, fast inter-connects, and **not** considering the time gains due to asynchronous updates, FullSync is about 20% slower than our method.*

| PP × DP Mesh | AWS Instances | Communication Time for tail (ms) | Forward-Backward time for tail (ms) | Speed-up |
|---|---|---|---|---|
| $4 \times 2$ | $1\times$ p4d.24 | $515 \pm 35$ | $320 \pm 25$ | $\sim 2.6\times$ |
| $4 \times 4$ | $1\times$ p4d.24 | $920 \pm 150$ | $590 \pm 35$ | $\sim 2.6\times$ |
| $4 \times 6$ | $2\times$ p4d.24 | $925 \pm 40$ | $580 \pm 15$ | $\sim 2.6\times$ |
| $4 \times 8$ | $2\times$ p4d.24 | $1240 \pm 70$ | $600 \pm 30$ | $\sim 3.1\times$ |
| $4 \times 12$ | $3\times$ p4d.24 | $1610 \pm 80$ | $595 \pm 25$ | $\sim 3.7\times$ |
| $2 \times 4$ | $1\times$ p4d.24 | $540 \pm 55$ | $470 \pm 35$ | $\sim 2.2\times$ |
| $4 \times 4$ | $1\times$ p4d.24 | $920 \pm 150$ | $590 \pm 35$ | $\sim 2.6\times$ |
| $6 \times 4$ | $2\times$ p4d.24 | $660 \pm 120$ | $480 \pm 75$ | $\sim 2.4\times$ |
| $8 \times 4$ | $2\times$ p4d.24 | $400 \pm 65$ | $815 \pm 35$ | $\sim 1.5\times$ |
| $12 \times 4$ | $3\times$ p4d.24 | $735 \pm 65$ | $355 \pm 85$ | $\sim 3.1\times$ |

Table 2: *We report the SPARTA communication time (i.e., averaging 5% of the parameters) for tail stage for the 8-layer base model, that would be masked by our asynchronous sparse averaging for various configurations above and compare it with the forward-backward times. This shows the empirical speed-up that could be obtained by our method, which ranges from $1.5\times - 3.7\times$ and improves with larger mesh. This speed-up only considers asynchronous DP and any benefits due to AsyncPP is additional to this.*

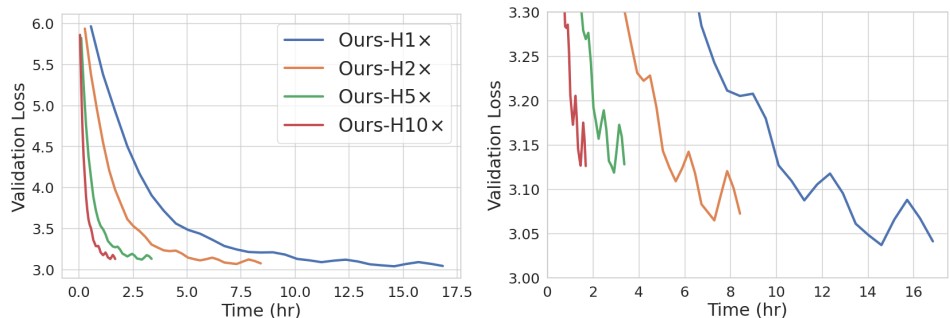

Figure 11: *Our method in a heterogeneous setup with drastically varying device speeds. The performance degradation is negligible compared to the gain in wall-clock time.*

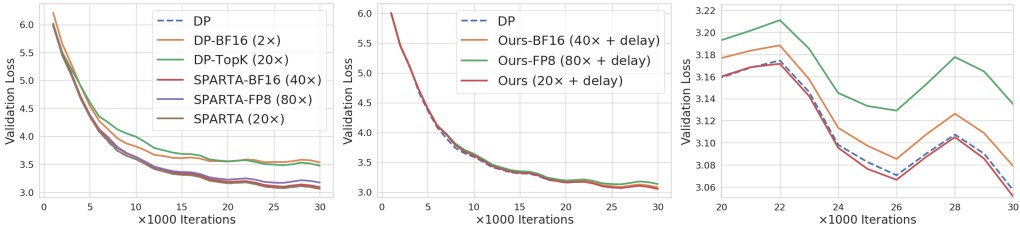

Figure 12: *Effect of compression for different methods, for the base model with $4 \times 2$ mesh on Wiki-Text.* **Left:** *Both quantisation and TopK sampling based on weight magnitude (instead of random) degrade the performance for* DP *(*DP*-FP8 did not converge). However, SPARTA is robust to quantization.* **Right:** *Similar to SPARTA, our method is robust to quantization even with a 10-step delay, and the degradation is minimal. TopK sampling did not converge for our method, aligning with our insight that the sampling needs to be unbiased.* DP *is even more sensitive to quantization with delay, and* DP*-BF16 with delay did not converge. The robustness of sparse averaging (even with delay) to quantization is intriguing, and it may be explained by the fact that since the quantization error is introduced only for a small subset (5% in this case) at each iteration, the effect of quantization on training is negligible. However, this warrants further study, which is beyond the scope of this work.*

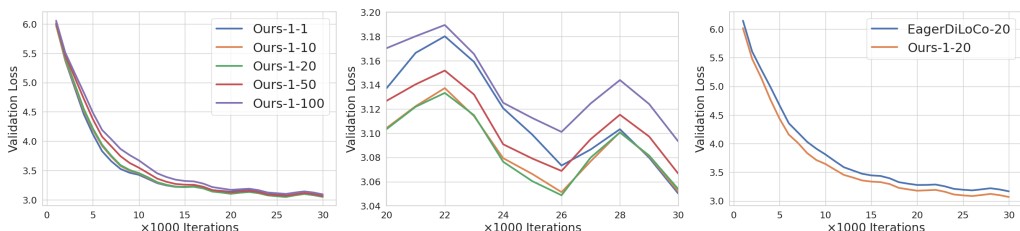

Figure 13: *Results in an equivalent setup to EagerDiLoCo (Kale et al., 2025). Ours-1-X denotes our method with subset size 1 (full averaging) and varying averaging interval (i.e., delay for asynchronous DP). On the right, we compare against EagerDiLoCo for 20 inner-steps (i.e., delay). Our EMA approach outperforms eager updates, confirming the strict generality, and the performance varies only slightly with different numbers of inner steps.*