# OpenReview forum: "AsyncMesh: Fully Asynchronous Optimization for Data and Pipeline Parallelism"
_ICLR.cc/2026/Conference — Submitted to ICLR 2026_

### Official Review · Reviewer_tRZk · 2025-10-26

**Soundness:** 3
**Presentation:** 3
**Contribution:** 3
**Rating:** 4
**Confidence:** 3

**Summary:**

The paper introduces a new distributed training framework called AsyncMesh, which enables fully asynchronous optimization across both data parallel (DP) and pipeline parallel (PP) axes. To counteract the staleness introduced by asynchrony, the authors propose (1) a weight look-ahead mechanism for PP using Nesterov extrapolation and (2) an asynchronous sparse averaging scheme for DP with an exponential moving average (EMA)-based staleness correction. They theoretically prove convergence under standard assumptions and empirically show that AsyncMesh achieves performance comparable to fully synchronous training while drastically reducing communication overhead.

**Strengths:**

+ Introduces AsyncMesh, that enables asynchronous updates across both data parallelism (DP) and pipeline parallelism (PP) to address this communication bottleneck.
+ Combines Nesterov-based weight look-ahead for PP and Exponential Moving Average (EMA) correction for DP to counteract stale gradients and parameters effectively.
+ Provides formal convergence guarantees for both asynchronous sparse averaging and delayed updates, extending existing results from stochastic approximation theory.
+ Demonstrates performance parity with fully synchronous baselines on language models up to 1 billion parameters, showing scalability and robustness.

**Weaknesses:**

- Theoretical convergence guarantees rely on homogeneous settings, which may not hold in practical heterogeneous or real-world decentralized systems.
- Although sparse averaging reduces communication, it could slow convergence for extremely small subsets or large delays, as hinted in the theoretical analysis. No experiments have done on this.
- The paper lacks direct comparisons with strong recent baselines such as DeepSpeed ZeRO, ZeRO++.
- The effects of EMA decay rates, subset sizes, and delay parameters are not deeply analyzed.
- Combining asynchronous DP and PP with custom staleness correction mechanisms may complicate integration into existing distributed training frameworks.

**Questions:**

1. The theoretical convergence proofs assume identical hardware, learning rates, and i.i.d. data across replicas. Extend the theoretical framework or provide an empirical ablation to quantify performance under heterogeneity, such as uneven compute power, data imbalance, or non-uniform network latency.
2. Could staleness correction via EMA still approximate global consensus effectively when replicas diverge in data distribution or update frequency? Discuss convergence guarantees in heterogeneous environments.
3. The paper claims sparse averaging (e.g., 5%) maintains performance, but what happens at extreme sparsity (e.g., 1% or less) or high delay (τ > 50)? Is there a theoretical threshold or empirical tipping point where sparse averaging starts to degrade convergence or stability? Conduct controlled experiments varying both subset size (1–10%) and delay intervals (10–100 steps) to observe convergence degradation patterns. Include convergence curves and communication–accuracy trade-off plots to better illustrate the regime where AsyncMesh remains stable.
3. Why were DeepSpeed ZeRO and ZeRO++ omitted from the baseline comparisons, given their dominance in large-scale model parallelism? How would AsyncMesh compare to ZeRO’s optimizer and gradient state partitioning in terms of both communication efficiency and memory footprint? Include at least one benchmark comparison (even partial) against ZeRO or ZeRO++ under similar mesh sizes to demonstrate where AsyncMesh provides unique advantages. Discuss interoperability potential — e.g., could AsyncMesh be layered on top of ZeRO’s optimizer partitioning?
4. How sensitive is AsyncMesh’s convergence to the EMA coefficient schedule (λₜ)? Does the same schedule generalize across datasets, model sizes, or communication delays? Are there interactions between subset size, EMA decay, and delay that affect stability?

---

> ### Author Response · Authors · 2025-11-17
>
> We thank the reviewer for recognizing the contributions and the constructive feedback. Below, we address the main concerns.
>
> ## (W1, Q1, Q2) Heterogeneous setting
> - We have **already provided empirical results in the heterogeneous setting** in Appendix B.2, Fig. 11. In short, our method can handle up to *10x difference in device speeds* with negligible performance degradation. Supporting heterogeneity in itself is a research question, and a method specifically designed for a heterogeneous setup would likely improve the current results.
> - Providing theoretical analysis in a simplified setting is a standard practice in machine learning literature. Following this, we have provided convergence proofs with the homogeneous assumption – **this itself is useful as our AsyncMesh setup is new**. To extend our analysis to the heterogeneous setting, we need to make additional assumptions to bound the weight deviations between replicas, similar to [1,2]. We believe this constitutes an interesting future direction.
>
> ## (W2, W4, Q1, Q3) Results with small subsets and large delays
> - We have **already provided results with varying subset sizes and delays in Sec. 5.3 Fig. 3.** As noted, our method is stable even with 1% subset size and up to a 50-step delay. These results clearly validate the empirical robustness of our approach.
>
> ## (W3, Q4) Comparison with ZERO and ZERO++
> - Frameworks like ZERO **enhance the system engineering aspect of DP** to improve memory efficiency, allowing large models to be trained on a small memory GPU cluster. These methods trade off communication for memory efficiency; thus, **they are not suitable for bandwidth-limited decentralized training scenarios**. In contrast, our focus is on improving communication efficiency (not memory efficiency) with asynchronous updates to enable decentralized training over bandwidth-constrained interconnects. Therefore, **these frameworks are orthogonal to our approach**, and it is possible to modify these frameworks to handle asynchronous PP/DP communication by utilizing ideas from our method. We will add a discussion and cite these papers.
> - Our main focus is to show that asynchronous DP updates do not degrade convergence with respect to the number of iterations. Since ZERO and ZERO++ only improve the implementation efficiency, and their convergence with respect to iteration is the same as DP, we believe the curves would overlap with that of DP, and therefore, it is not clear if they would add value.
> - Similar to ZERO++, we have tested our asynchronous sparse averaging with quantization to further improve the communication efficiency. These results are provided in Appendix B.2 Fig. 12. Clearly, **our method is robust even with FP8 quantisation**.
>
> ## (W4, Q5) Sensitivity of EMA coefficient
> - We found that the EMA coefficient $\lambda_t$ has minimal impact on convergence (Fig. 5) and is robust across various settings. Therefore, we kept it **constant for all experiments** in the paper.
>
> ## (W5) Implementing asynchronous updates into existing frameworks
> - We believe that if there are benefits as observed in our experiments, it would be worth the effort to incorporate asynchronous methods in existing distributed training frameworks. We consider this **alternate approach** as a research question, and as asynchrony completely overlaps communication and computation, it may even simplify the over-engineering required to improve device utilization in synchronized training frameworks. For example, one can use AsyncPP instead of ZeroBubble [3] to improve pipeline utilization.
>
> ## References
> - [1] Wang, Jiayi, et al. "A new theoretical perspective on data heterogeneity in federated optimization." arXiv preprint arXiv:2407.15567 (2024).
> - [2] Lee, Sunwoo, et al. "Partial model averaging in federated learning: Performance guarantees and benefits." Neurocomputing 556 (2023): 126647.
> - [3] Qi, Penghui, et al. "Zero bubble pipeline parallelism." arXiv preprint arXiv:2401.10241 (2023).

---

> > ### Comment · Reviewer_tRZk · 2025-11-22
> >
> > I thank the authors for addressing my previous comments. However, several key concerns remain insufficiently addressed:
> >
> > 1.	Heterogeneity:
> > The rebuttal only examines heterogeneous compute speed, while my questions focused on non-i.i.d. data, uneven update frequencies, and heterogeneous network delays—core challenges in decentralized training. The theoretical analysis still assumes full homogeneity, and no additional empirical results are provided. Thus, applicability to realistic heterogeneous settings remains unclear.
> >
> > 2.	Extreme sparsity and large delays:
> > My request explicitly asked for exploring higher delays (τ>50) and identifying degradation thresholds or failure modes. No new experiments or analysis are provided.
> >
> > 3.	Missing ZeRO / ZeRO++ comparisons:
> > Stating that ZeRO is “orthogonal” does not replace the need for at least a small-scale comparison or communication/memory analysis. ZeRO includes communication optimizations and is a standard baseline for large-scale distributed training. The rebuttal provides no empirical or quantitative comparison.
> >
> > 4.	EMA sensitivity:
> > The response relies on a single ablation and does not address interactions with larger delays, smaller subsets, or multiple model sizes.
> >
> > 5.	Integration complexity:
> > The rebuttal offers a high-level opinion but no concrete analysis or evidence regarding practical integration challenges.
> >
> > Overall, while I appreciate the clarifications, the rebuttal does not fully resolve the major concerns, and my evaluation of the paper remains unchanged.

---

> > > ### Author Response · Authors · 2025-12-01
> > >
> > > We thank the reviewer for their engagement during the discussion period.
> > >
> > > ## Heterogeneity
> > > We agree with the reviewer that our main focus is the homogeneous setup, and the heterogeneous device-speed results are included as a *curiosity-driven experiment* to see how our method would perform if any heterogeneity is introduced. As the reviewer mentioned, studying heterogeneity across various aspects (such as data and device/network-speed) is important; however, we believe it would significantly expand the scope of this work, and a rigorous theoretical and empirical study of heterogeneity would constitute a paper in itself.
> > >
> > > To support our argument, just considering the theoretical literature, **data heterogeneity, and device/network heterogeneity are typically studied separately** [1,3] due to their distinct nature. Furthermore, asynchronous methods in just the homogeneous setup have been published in prestigious venues [2], and extending it to ONE of the heterogeneous setups (data or compute speed) is typically studied separately [3]. Also, theoretical papers do not typically include large-scale experiments. In our work, we have shown 1B parameter-scale LLM results along with theoretical analysis in the homogeneous setup, and we kindly request the reviewer to assess the merits of the paper while considering this context.
> > >
> > > Additionally, we would like to clarify that our setup assumes an i.i.d. data setup (Sec. 2.1), which is a **realistic assumption for decentralized training scenarios**, in contrast to the federated learning setup, which typically considers data heterogeneity.
> > >
> > > ## Extreme sparsity and delays
> > > We thank the reviewer for requesting more fine-grained empirical analysis of our method with respect to the subset size and the async delay. As per the reviewer’s request, we have run experiments with a 0.005 subset size and 60, 70, and 90-step delays, in addition to the already provided results in Fig. 3 (subset sizes: [ 0.1,  0.05, 0.02, 0.01, 0.001], and async delays: [100, 50, 20, 10, 1]).
> > > - With the subset size 0.005, the validation perplexity gap is only **0.43**, and this increases to 1.59 when we reduce the subset size to 0.001.
> > > - With respect to async delays, our method has a distinct behaviour, which is that *beyond a 60-step delay, our method did not converge*. Specifically, with a 60-step delay, the perplexity gap is **0.16**; however, the method starts to diverge after 3000 training steps for the experiments with delays 70, 90, and 100. We believe the delay is too large for the EMA to accurately estimate the expected staleness as per our Theorem 2 for this model configuration.
> > >
> > > ## ZERO/ZERO++
> > > Asynchronous methods **eliminate the communication overhead by construction**, as they completely overlap communication with computation, *in contrast to the ZERO-type methods, which improve the implementation efficiency via system engineering techniques.* Since **these two aspects are complementary**, our implementation could also be improved via system-engineering enhancements similar to ZERO or related methods. Furthermore, as mentioned previously, ZERO-type methods are communication-heavy and **NOT applicable in the bandwidth-constrained scenarios**, which is our main focus. Therefore, we are unclear what one can gain from comparing with ZERO. While we are open to doing more experiments that would better clarify our method, we are unclear about the reason and therefore unable to prioritise this during the short rebuttal period.
> > >
> > > ## EMA sensitivity
> > > We would like to clarify that **all experiments (base model, 1B model, varying subset size, delays, mesh sizes, heterogeneous setup) in our paper use the same EMA coefficient**. While one may be able to tune this EMA coefficient for each setup, being able to use the same coefficient clearly demonstrates that **the method is less sensitive** to it. If the reviewer can clarify what *interactions* they are looking for, we can try to do that experiment.
> > >
> > > ## Integration complexity
> > > We believe that if asynchronous methods become popular or useful, they will be integrated into existing distributed systems. We will even argue that **even if it is not trivial to integrate asynchronous methods into the existing systems, that must NOT stop researchers from exploring asynchronous methods**. Having said this, the open source collaborative distributed training frameworks such as Hivemind [4] are more amenable to asynchronous methods as their communication primitives are peer-to-peer without stringent global synchronization requirements. We kindly request the reviewer to have an open mind about alternative training approaches that may not be easily integrated into existing systems.

---

> > > > ### Author Response · Authors · 2025-12-01
> > > > **References**
> > > >
> > > > - [1] Wang, et al. "A new theoretical perspective on data heterogeneity in federated optimization.", ICML 2024
> > > > - [2] Maranjyan, et al. "Ringmaster ASGD: The first Asynchronous SGD with optimal time complexity." ICML 2025.
> > > > - [3] Ringleader ASGD: The First Asynchronous SGD with Optimal Time Complexity under Data Heterogeneity, arXiv:2509.22860, 2025.
> > > > - [4] Ryabinin, Max, et al. "Hivemind: Decentralized Deep Learning in PyTorch, April 2020." URL https://github. com/learning-at-home/hivemind.

---

### Official Review · Reviewer_yc1x · 2025-10-28

**Soundness:** 1
**Presentation:** 2
**Contribution:** 1
**Rating:** 2
**Confidence:** 4

**Summary:**

This paper try to incorporate both async communication in DP(data parallel) and PP(pipeline parallel).

The major contribution is to combine asyncPP and SPARTA in DP together and guarantee the loss curve can converge.

**Strengths:**

1. Theoretical analysis on AsyncPP and SPARTA in DP

2. e2e experiments training and show loss curves

**Weaknesses:**

1. the major experimental model is a toy size of 160M, which cannot represent real world pre-training model patterns. In addition, it is just toy NanoGPT not a real GPT model. Furthermore, the model does not even have basic dropout layer, which make the loss curve comparison less convincing.

2. the paper contribution is very minor, it just combined existed work AsyncPP and SPARTA in DP together and did a bit tuning. There is very little research novelty here.

3. Whether the model can converge or not in such async model training case is heavily depend on delayed steps, the math part of this paper does not even discuss much about it thus making the whole proof less meaningful.

4. the sec 5.4, 1B model itself is not a standard GPT model, for example, the embedding dim is very small as 2304, and 24 attention head is not standard gpt-3. In addition, the paper does not show any main stream and standard model size results, thus making the result less convincing.

5. the authors lack of knowledge about SOTA data parallel framework, such as ZeRO or FSDP, which is the only DP paradigm used in real world. And people start only use them as DP starting from 2020. But there is no discussion on how to do async communication overhead in such schemes.

**Questions:**

If t - $\tau$ to t has a big delay, how should loss converge still hold? The proof does not make sense if there is a big gap between t - $\tau$ to t. And there is no discussion on how to theoretically analyse and determine the biggest gap between t - $\tau$ to t to make the loss curve difference minimal to fully synced.

---

> ### Author Response · Authors · 2025-11-17
>
> We thank the reviewer for the constructive feedback. Below, we address the main concerns.
>
> ## (W1, W4) Model configuration
> - We use NanoGPT as it is a popular repository for research on **GPT2-style models**. Our base model is larger than GPT2-base (124M vs 160M parameters), and our 1B model has a larger embedding dimension than GPT2-xl (1600 vs 2304) with a similar number of attention heads (25 vs 24). Therefore, we respectfully disagree with the statement that “our model is toy and the embedding dimension of our 1B model is very small and 24 attention heads is not standard.”
> - We disabled dropout as it did not make a noticeable difference in our early experiments.  Furthermore, we would like to stress that **dropout is not a standard component** in the latest state-of-the-art architectures (eg., Gemma-3, Grok-1, OLMO-2, to name a few).
> - Our model configuration choices are driven by the feasibility of experiments given our compute resources, and the configuration is close to the GPT2 architecture. Therefore, **we do not believe this model configuration would greatly benefit one method over the other.** We note that the 160M base model allowed us to test up to 12 x 4 and 4 x 12 PP x DP mesh sizes (Fig. 3), and the 1B model shows that the conclusions of the base model hold at the billion-parameter scale (Fig. 4).
> - We would like to recalibrate the expectation that small research groups do not have the computing resources that large industry labs do, and therefore, it is not possible to pretrain multi-billion parameter scale models with our available compute resources. However, if the reviewer has any model configuration in mind that would change their view about the results, we are more than happy to do the experiment.
>
> ## (W2) Novelty, Contribution, and Significance
> - **(Novelty)** Our AsyncMesh setup is novel, where we consider **asynchronous updates for both Pipeline and Data Parallel axes**. Previously asynchronous updates are *only* considered for PP or DP *individually*, and the training behaviour is unclear when both are allowed to be asynchronous – this paper investigates this setup.
> - **(Contribution)** For AsyncPP, we adopt the method recently published in ICML (Ajanthan et al., 2025a), and our main methodological contribution is making stagewise DP asynchronous – Section 3 focuses on this. Our final approach of estimating average staleness via EMA to correct delay is new, and we combine it with sparse averaging to reduce data transfer volume. In addition to proving **theoretical convergence**, we show **superior empirical results** with up to 1B parameters (Fig. 4), and robustness across varying subset sizes, staleness levels, DP communication intervals, and PP × DP mesh sizes in Fig. 3.
> - **(Significance)** In retrospect, given the simplicity and effectiveness of our method, one might overlook the complexity of the AsyncMesh setup. We would like to emphasize that asynchronous optimization is significantly more challenging than synchronous optimization due to gradient staleness, which is more severe when both PP and DP are asynchronous (two sources of staleness), as in our case. And this is the first time such a setup is investigated in a real-world LLM training (up to 1B parameters), showing **no performance degradation** compared to the FullSync method, along with a convergence proof. Since AsyncMesh eliminates the communication overhead in large-scale distributed training in both PP and DP dimensions, we believe it constitutes a new, communication-efficient way of training foundational models. Therefore, we respectfully disagree with the assessment that “the contribution is very minor”.
>
> ## (W3, Q1) Delay and convergence
> - This is a good observation that the delay is not explicit in the theoretical convergence statement. In fact, the delay $\tau$ is implicit in the assumptions of Theorem 2 that the drift in staleness is diminishing $\|\alpha_t\|\to 0$ and $\sum_t \|\alpha_t\|^2/\lambda_t < \infty$. This assumption will be valid only if the *optimization trajectory is smooth and the delay is small*, as the EMA should be able to estimate the time-varying staleness $D^t$ as mentioned in lines 293 - 301. We will make this clear in the revised version.
> We show the practical effects of delay in Fig. 3, showing that our method is robust up to 50-step delay.

---

> > ### Author Response · Authors · 2025-11-17
> >
> > ## (W5) SOTA DP frameworks
> > - Frameworks like ZERO and FSDP **enhance the system engineering aspect of DP** to improve memory efficiency, allowing large models to be trained on a small memory GPU cluster. These methods trade off communication for memory efficiency; thus, **they are not suitable for bandwidth-limited decentralized training scenarios**. In contrast, our focus is on improving communication efficiency (not memory efficiency) with asynchronous updates to enable decentralized training over bandwidth-constrained interconnects. Therefore, **these frameworks are orthogonal to our approach**, and it is possible to modify these frameworks to handle asynchronous PP/DP communication by utilizing ideas from our method. We will add a discussion and cite these papers.
> > - Our main focus is to show that asynchronous DP updates do not degrade convergence with respect to the number of iterations. Since ZERO and FSDP only improve the implementation efficiency, and their convergence with respect to iteration is the same as DP, we believe the curves would overlap with that of DP, and therefore, it is not clear if they would add value.

---

### Official Review · Reviewer_uqSG · 2025-11-01

**Soundness:** 3
**Presentation:** 3
**Contribution:** 2
**Rating:** 4
**Confidence:** 2

**Summary:**

This paper proposes AsyncMesh, asynchronous staleness-aware training approach that combines an asynchronous sparse averaging method and an exponential moving average based correction mechanism. It also provides convergence guarantees for both sparse averaging and asynchronous updates and evaluates their methods using LLMs with up to 1B parameters.

**Strengths:**

1. AsyncMesh explores the setup where both DP and PP are asynchronous.
2. The paper designs an Exponential Moving Average (EMA) based correction mechanism that approximates the average staleness.
3. The paper provides theoretical justification of convergence in the presence of staleness in a homogeneous setup where only a small subset of weights is communicated between DP replicas.

**Weaknesses:**

1. The baseline for the evaluation is weak. The benchmark for this evaluation is weak. The evaluation only compares AsyncMesh with FullyAsync and DP. However, well-studied staleness-aware LLM training [1]  (with different degree of staleness) and also block coordinate descent with correction [2] was not included in the evaluation.
2. The evaluation results did not show how much performance improvement sparse averaging could bring.

[1] PipeDream: Generalized pipeline parallelism for DNN training
[2] Accelerating Block Coordinate Descent for LLM Finetuning via Landscape Correction

**Questions:**

Does the A00 machine used in the evaluation have NVLink?

---

> ### Author Response · Authors · 2025-11-17
>
> We thank the reviewer for recognizing the contributions and constructive feedback. Below, we address the main concerns.
>
> ## (W1) Evaluation baselines
> - Our objective is to show how different Data Parallel (DP) methods perform when combined with asynchronous optimization for Pipeline Parallel (AsyncPP). Therefore, except FullSync, all methods use AsyncPP for PP optimization as mentioned in the captions of Fig. 2 and Table. 1. As shown, **our method (while both PP and DP are asynchronous) matches the fully synchronous method (FullSync)**.
> - AsyncPP ICML paper (Ajanthan et al., 2025a) clearly showed that it is significantly better than PipeDream [1] due to staleness correction with the Nesterov method; therefore, we believe comparison with [1] in this paper does not add value.
> - [2] is a block coordinate descent method designed to reduce the memory requirement of optimizer states for LLM fine-tuning. Since our focus is **pretraining** and especially with **asynchronous optimization** in the PP x DP mesh, it is unclear if [2] is applicable in our scenario.
> - If the reviewer suggests any directly relevant baselines that are not compared, we are happy to do that experiment during the rebuttal period.
>
> ## (W2) Improvement due to sparse averaging
> - The main results show that even with asynchronous sparse averaging, the results match FullSync. Since only 5% of the parameters are averaged, sparse averaging **reduces the communication requirement by 10x** (subset and its indices need to be communicated). Furthermore, by making it asynchronous, we further improve the speed-up by **1.5x - 3.7x** on top of synchronous sparse averaging. This is briefly discussed in the paragraph starting at line 443, and the detailed results and discussion are provided in Appendix B.2 and Table 2.
>
> ## (Q1) A100 machine
> - We use the standard p4.24d instance in AWS, which has NVLink. Since we control the delay (set to a 10-step delay by default), inter-node bandwidth does not affect the reported results.

---

> > ### Comment · Reviewer_uqSG · 2025-11-26
> >
> > Dear author,
> >
> > Thank you for your reply. Based on the more detailed information you provided, this paper has one more weakeness accroding to my opinion.
> >
> > Since the experiments were conducted on instances equipped with 8 A100 GPUs and NVLink, a good training parallelism plan should include tensor parallelism in most cases. However, the AsyncMesh setup only includes pipeline parallelism and data parallelism. Therefore, these experiments cannot help demonstrate the performance improvements that AsyncMesh can provide in popular LLM training configurations.
> >
> > I recommend that the authors use the popular LLM training configurations when comparing AsyncMesh with other baselines.

---

> > > ### Author Response · Authors · 2025-12-01
> > > **TP is not suitable for bandwidth-constrained scenarios**
> > >
> > > We thank the reviewer for their engagement during the discussion period.
> > >
> > > **Popular parallelism dimensions are DP and PP**, and Tensor Parallelims (TP) is rarely used. Specifically, TP would only be used in cases where a single layer of a model cannot fit into a GPU, and it is the **most communication-inefficient** parallelism strategy. Therefore, *it is NOT suitable for bandwidth-constrained scenarios, as in our setup*. We kindly refer the reviewer to https://huggingface.co/spaces/nanotron/ultrascale-playbook for a detailed overview of the parallelism strategies used in practice and their pros and cons.

---

### Author Response · Authors · 2025-11-19
**Illustrative video to clarify the AsyncMesh setup**

To clarify the AsyncMesh setup and address potential misunderstandings, we have included a short **7-minute illustrative video** in the supplementary material. The video contrasts our approach with the standard synchronous setup and highlights the novelty, associated challenges due to asynchronous updates, and how we mitigate them. We kindly encourage reviewers to view it for additional context.

---

### Meta-Review · Area_Chair_oVbD · 2026-01-06

**Summary:**

The manuscript proposes AsyncMesh, combining asynchrony along both data-parallel and pipeline/model-parallel axes with staleness-mitigation, and reports experiments up to 1B parameters alongside convergence analysis. Two recent asynchronous methods introduced for both of the above are utilized with some adaptations to make them work well. A main contribution of the paper is to show that these can lead to no performance degradation in a specific bandwith constrained setting leading to faster training in that setting. Reviewers and the meta-reviewer appreciated the goal and aspects of the technical approach and the empirical result, but some concerns remain on baselines and clarifying and emphasizing the novelty. Overall the authors are encouraged to revise and resubmit the manuscript.

**Reviewer Concerns:**

(1) Comparisons to existing baselines, although this was partially addressed by pointing out the background papers show SOTA it seems important to include prior methods.

(2) novelty - the paper combines two existing approaches, the methodological contributions (such as the novel adaptations to make them work together) and the significance of the empirical results can be better emphasized in the manuscript.

(3) ZERO/FSDP and TP are popular and widely used schemes often used in place or in conjunction with PP depending on the bandwith constraints and that have throughput benefits (compared to PP) in high bandwidth settings. These are not completely orthogonal as they all address ways of fitting large model training on multiple accelerators that cannot fit the entire replica. This was only partially adressed and should be more clearly addressed in the manuscript. The authors are advised to revisit some of their claims.

**Reviewer Scores:**

Speculatively, trZK had a number of ablations and concerns well addressed and would likely raise their score. The other reviewers keep their scores.

---

### Decision · Program_Chairs · 2026-01-26

Reject